# Antimicrobial Preservatives for Protein and Peptide Formulations: An Overview

**DOI:** 10.3390/pharmaceutics15020563

**Published:** 2023-02-07

**Authors:** Luisa Stroppel, Torsten Schultz-Fademrecht, Martin Cebulla, Michaela Blech, Richard J. Marhöfer, Paul M. Selzer, Patrick Garidel

**Affiliations:** 1Innovation Unit, PDB-TIP, Boehringer Ingelheim Pharma GmbH & Co. KG, Birkendorfer Straße 65, 88397 Biberach an der Riss, Germany; 2Interfaculty Institute of Biochemistry, Eberhard Karls Universität Tübingen, Auf der Morgenstelle 34, 72076 Tübingen, Germany; 3Boehringer Ingelheim Animal Health, Binger Str. 173, 55216 Ingelheim am Rhein, Germany

**Keywords:** antimicrobial preservative, protein formulation, multi-dose, *m*-cresol, phenol, benzyl alcohol, chlorobutanol, methylparaben, propylparaben, paraben, benzalkonium chloride, phenoxyethanol, preservative protein interaction

## Abstract

Biological drugs intended for multi-dose application require the presence of antimicrobial preservatives to avoid microbial growth. As the presence of certain preservatives has been reported to increase protein and peptide particle formation, it is essential to choose a preservative compatible with the active pharmaceutical ingredient in addition to its preservation function. Thus, this review describes the current status of the use of antimicrobial preservatives in biologic formulations considering (i) appropriate preservatives for protein and peptide formulations, (ii) their physico-chemical properties, (iii) their in-/compatibilities with other excipients or packaging material, and (iv) their interactions with the biological compound. Further, (v) we present an overview of licensed protein and peptide formulations.

## 1. Introduction

The number of biological formulations has been rising over the last years and will continue to increase in the future [1,2]. The therapeutic indication as well as the patient needs require to define the appropriate formulation, dosage and delivery system as stated in the target drug product profile. Thus, the target drug product profile outlines the desired characteristics of a target drug product that is aimed for a particular disease. It summarizes the intended use, target populations, and other desired attributes of products, including safety and efficacy-related characteristics, as well as drug delivery system. Developing a new formulation, scientists have to decide whether the drug should and can be applied via single-dose or multi-dose administration. Even though multi-dose formulations contributed to one third of marketed protein pharmaceuticals as already described in 2007 [3], their usage is limited to a small number of active pharmaceutical ingredients (API), mostly insulin [4]. Nevertheless, multi-dose formulations present several advantages. First, they provide a better dosage management and therefore, minimize product wastage [3,5]. Second, the drug administration is facilitated for the patient or the health staff since the same container can be used over several days or weeks, also providing a higher dosage flexibility [5]. Third, they are less cost-intensive in terms of the production process. For instance, it was calculated that the production of a 10-dose vial is 2.5× cheaper compared to a single dose vial, including factors, such as filling costs, overfill per dose, and packaging costs [6,7]. Last, less packaging waste is produced which has not only economic advantages, but sustainability aspects as well [3].

As described in the pharmacopoeia, multi-dose drugs must show antimicrobial effectiveness against the growth of a broad spectrum of microorganism [8,9,10]. Bacteria, yeasts, and fungi have different metabolisms and can grow in drug products if the environmental conditions are suitable and if nutrients are available. Indeed, almost all pharmaceutical formulations provide the sufficient amount of nutrients as most ingredients are biodegradable. For example, Favero et al. (1971) reported that even distilled water contained enough nutrients to enable the growth of *pseudomonas* [11]. L-arginine and sucrose are additional examples of excipients favoring germination [12,13]. Moreover, factors, e.g., pH and temperature, have a considerable impact on microbial growth [12,13]. If the formulation does not provide adequate intrinsic protection against microbes, antimicrobial preservatives must be present in the formulation [1,12,14]. However, several reports showed that preservatives increase protein and peptide destabilization, such as particle formation (in the following, proteins and peptides are summarized under the term “biological compounds“ [5,15,16,17,18,19,20,21,22,23]. Thus, it is crucial to be aware of the preservative’s properties and its effect on the stability of a biological compound.

Taken together, this review provides an overview of the most commonly used antimicrobial preservatives (AP) for biological formulations. It serves as a guideline for choosing an appropriate AP with regard to parameters, e.g., antimicrobial efficacy, physico-chemical formulation properties, incompatibilities, interactions with biological compounds, publicly available and published experience and case studies using such APs with biological compounds, and products currently on the market.

## 2. Selection of an Antimicrobial Preservative

The current chapter discusses selection criteria for choosing a preservative system intended for a multi-dose formulation. As described in the literature, suitable preservatives for biological formulations are *m*-cresol (CR), phenol (PH), benzyl alcohol (BA), benzalkonium chloride (BAK), chlorobutanol (CB), phenoxyethanol (PE), methylparaben (MP), propylparaben (PP), or mixtures thereof [3,5]. Although phenoxyethanol has only been used in vaccines so far, it has shown less destabilizing effects on peptide, protein, and antibody stability compared to other preservatives making it a promising antimicrobial candidate [15,17,22]. Methylparaben and propylparaben are often used in combination mostly at a concentration of 0.18% w·v^−1^ (11.8 mM) and 0.02% w·v^−1^ (1.1 mM), respectively [12,24,25]. Combinations are used as the antimicrobial activity of parabens increases with increasing carbon chain length while, concomitantly, a longer carbon chain decreases the solubility [5,12,14,26]. For instance, the mixture of methyl- and propylparaben is used in the formulation of Maxitrol (Sandoz International GmbH, Basel, Switzerland) or Syntocinon (Novartis-AG, Basel, Switzerland). The use of organomercurial preservatives, e.g., thiomersal, has declined over the years due to toxicity concerns [13,24,27]. Thus, these preservatives are not discussed in this review.

Preservatives can be categorized regarding their structural composition and antimicrobial activity. Phenol and *m*-cresol are phenolic derivatives being active against Gram-positive and -negative bacteria, but not against spores [24,28]. In general, spores are more resistant against antimicrobial agents due to their structure and composition. They have an unique anatomy containing different layers of concentric protein shells. Starting from the outside, the first shell is the exosporium followed by the coat and the core. The inner structures are very much the same between species, but the outer, the exosporium, if present, can significantly differ. On the one side, these layers serve as a permeability barrier, and on the other side, contain enzymes which are able to detoxify reactive agents [29,30]. A detailed review about the resistance of bacterial spores to chemical agents is given by Setlow (2013) [13]. Benzyl alcohol, chlorobutanol, and phenoxyethanol belong to the class of alcohols. This class exhibits rapid antimicrobial efficacy against a broad range of vegetative bacteria, viruses, and fungi but is not active against spores [28,31]. Methylparaben and propylparaben are classified as benzoic acid derivatives. They are active against a broad spectrum of microorganisms and most efficient against molds and yeasts [14]. Last, benzalkonium chloride belongs to the class of quaternary ammonium compounds (QAC) which are referred to as cationic agents [12,28]. This class of preservatives is positively charged and has amphiphilic properties. In particular, benzalkonium chloride is defined as a mixture of alkylbenzyldimethylammonium chlorides with the general formula [C_6_H_5_CH_2_N(CH_3_)_2_R]Cl with R standing for a mixture of different alkyls ranging from dodecyl to hexadecyl residues; thus, only an average molecular weight (MW) of 360 g·mol^−1^ can be determined [14]. QACs are primarily active against Gram-positive bacteria. Higher preservative concentrations are required for Gram-negative bacteria, and *Pseudomonas aeruginosa (P. aeruginosa)* seems to be highly resistant. Moreover, QACs have antifungal properties [13].

Additionally, the preservatives differ in their octanol:water partition coefficient (log*P*); the log*P* value describes the ratio in which a substance is distributed in the water and octanol phaseproviding information about a substance’s lipophilicity vs. hydrophilicity [32]. This parameter is important as a correlation was shown between a preservative’s hydrophobicity and its effect on the stability of the biological compound with more hydrophobic preservatives causing greater protein destabilization [5,15,19,33]. For example, the ranking order in which preservatives were bound onto a model peptide correlated with the octanol:water partition coefficient with greater values leading to increased binding behavior (CR (log*P:* 1.98) > PH (log*P:* 1.48) > BA (log*P:* 1.05)) [19]. The same log*P* ranking was reported in a study analyzing the effect of preservatives on a monoclonal IgG1 antibody [15]. More details are found in Section 3: Interactions between antimicrobial preservatives and proteins and peptides. In Table 1, the structural formulae, molecular weight, and log*P* values of the preservatives are summarized.

The efficacy of an AP is determined by intrinsic and extrinsic factors [12,34]. Intrinsic factors are inherent to microorganisms and defined as a microorganism’s ability to be resistant against a preservative and/or to prevent a preservative’s activity. They include the nature, composition (natural, chromosomally controlled property), and condition of the microorganism [12,34,35]. For example, Gram-negative bacteria have an additional cell wall which is not present in Gram-positive bacteria making them less vulnerable against antimicrobial agents [35,36]. Mycobacteria do also show a high resistance against a wide range of antimicrobial agents, e.g., quaternary ammonium compounds, such as benzalkonium chloride [28]. This phenomenon is attributed to the high lipid content in their cell wall, especially the presence of mycolic acid, making the cell wall very hydrophobic and resistant against preservatives [34]. Antimicrobial activity against mycobacteria was reported for some preservatives, e.g., phenol [14]. Last, microorganisms can “develop” resistance by degrading the preservative. For instance, parabens were shown to be degraded by an esterase of *Pseudomonas cepacian* [37]. The selection of an appropriate preservative includes external factors. External factors cover parameters, e.g., the preservative’s concentration, the pH of the formulation, the used packaging material and storage conditions, as well as interactions with other formulation components [12,34]. External factors are discussed in more detail in the following subchapters. 

### 2.1. Antimicrobial Activity of Preservatives

Antimicrobial preservatives should guarantee sufficient protection against a broad spectrum of microorganisms. Some preservatives have the same targets, and thus can be classified into different groups (Table 2). For example, compounds can accumulate in the cell membrane, thereby disorganizing its structure, which can lead to a loss of metabolites and ions or a change in the electrical potential or pH [38]. Depending on their structural composition and concentration, they can interact with target sites on the microbial cell surface or inside the cell [13,28]. Most of the used preservatives have intracellular target sites and thus must penetrate the outer layer of the microbial cell. Consequently, the preservative’s efficiency is directly correlated with its ability to interfere with the cell membrane or cell wall [35]. In general, preservatives are more active against Gram-positive than Gram-negative bacteria. The antimicrobial efficiency is described in the minimum inhibitory concentration (MIC) which is defined as the lowest concentration required to inhibit the bacterial growth [34]. For instance, benzalkonium chloride is more active against Gram-positive bacteria than against Gram-negative bacteria and is ineffective against *P. aeruginosa* strains [14]. Other preservatives such as methylparaben show a higher activity against molds and yeasts compared to bacteria [14]. However, preservatives often have multiple target sites which makes it difficult to find the causes of microbial cell death [39]. A detailed overview of the mode of action of preservatives compatible with biological formulations is described in the following subchapters. 

#### 2.1.1. Phenolic Derivatives

Phenolic derivatives exhibit membrane-active properties, leading to cell lysis [28]. For example, Lambert & Hammond (1973) reported that phenolic derivatives induced the efflux of potassium ions out of the microbial cell leading to cell death [40]. Additionally, it was reported that the presence of phenol (320 µg·mL^−1^ (3.4 mM)) led to rapid cell lysis of different *Escherichia coli* (*E. coli*) strains, pneumococci, *Bacillus subtilis* (*B. subtilis*), and staphylococci. In agreement to Srivastava & Thompson (1966), it was suggested that the tested preservatives influence the separation of daughter cells, as young bacteria were more susceptible to phenol than older ones [41,42]. Kroll & Anagnostopoulus (1981) showed that the presence of 0.75% w·v^−1^ (79.9 mM) phenol induced increased deplasmolysis (influx of water) in *P. aeruginosa*, leading to a disrupted electrochemical gradient [43]. Furthermore, it is presumed that phenol induces protein denaturation and precipitation [44]. Dagley et al. (1950) studied the effect of phenol and different alcohols, e.g., ethanol, *n*-propanol, and *n*-butanol, on the lag phase of *Aerobacter aerogenes* in a glucose ammonium salt medium [45]. Bacterial growth is divided into four phases, starting with the lag phase. In this phase, the bacteria adopt to their environment and produce macromolecules required for cell division (log phase) [46]. Thus, they increase only in size but not in cell number. It was demonstrated that phenol at concentrations from 4.15 mM to 8.3 mM led to a prolonged lag phase, leading to an overall reduced growth rate. It was hypothesized that phenol interfered with the metabolism of amino acids required for cell division. Indeed, it was shown that the presence of certain amino acids and/or their metabolic products induces a lag phase (fumarate and malate) whereas other compounds are even able to abolish (α-ketoglutarate and succinate) it. Thus, Dagley et al. (1950) hypothesized that the mode of action of phenol was based on its ability to inhibit the production of essential amino acids as well as their conversion to other amino acids important for bacterial cell division [45].

#### 2.1.2. Alcohols

Benzyl alcohol was attributed with membrane-active properties leading to increased membrane fluidity and reduced structural stability. Additionally, it was shown by fluorescent spectroscopic assays that benzyl alcohol interferes with bacterial membrane properties, e.g., efflux pumps [47]. In a different study, Karabit et al. (1986) reported that benzyl alcohol (1%) had a greater antimicrobial activity against Gram-negative bacteria compared to Gram-positive bacteria or fungi [48]. This conclusion was drawn based on the decimal reduction time (D-value) which was calculated from the negative reciprocal slope of the viable organisms with smaller values standing for a higher antimicrobial effectiveness. Thus, the D-values served as an indicator of a microorganism’s resistance to the preservative. At pH 5.0, the D-value was 0.37 for *E. coli* (Gram-negative), 5.48 for *Staphylococcus aureus* (*S. aureus*) (Gram-positive), 39.0 for *Candida albicans* (*C. albicans*) (yeast), and 28.8 for *Aspergillus niger* (*A. niger*) (mold). Additionally, the activity of benzyl alcohol was pH dependent, being greater in acidic conditions. For example, the D-value for *A. niger* was 28.8 at pH 5, whereas it was significantly increased at pH 6 being 76.8 [48]. 

Phenoxyethanol was reported to uncouple oxidative phosphorylation from the respiratory chain by translocating protons across the membrane [49,50] and to inhibit malate dehydrogenase in microorganisms [50]. It was reported to be more efficacious against *E. coli* than against *P. aeruginosa,* indicating differences in the mode of actions [49,50].

#### 2.1.3. Quaternary Ammonium Compounds

Quaternary ammonium compounds, e.g., benzalkonium chloride, are positively charged. Thus, they are attracted by the negatively charged microbial outer layer, e.g., specific phospholipids and lipopolysaccharide [28]. Furthermore, hydrophobic interactions occur between their alkyl residues and the lipid bilayer leading to membrane penetration [44]. Richards et al. (1976) used electron microscopy to evaluate the effect of benzalkonium chloride on cultures of *P. aeruginosa* (Gram-negative bacteria) [51]. The cells were reported to peel off their outer cell membrane when grown in a nutrient medium including 50 and 100 µg·mL^−1^ (138.9 mM and 277.8 mM) benzalkonium chloride as well as on solid medium containing benzalkonium chloride up to a concentration of 400 µg·mL^−1^. The cellular morphology was not changed in the presence of the preservative. This finding led to the conclusion that benzalkonium chloride caused the removal of the outer cellular membrane but had no effect on the peptidoglycan layer of the Gram-negative bacterium. Additionally, antimicrobial activity increased when 23 µg·mL^−1^ sodium edetate was combined with 500 µg·mL^−1^ benzalkonium chloride compared to the activity when used individually [51].

#### 2.1.4. Parabens

On a macroscopic level, the activity of parabens is attributed to their ability to disrupt the lipid bilayer and thus lead to an altered membrane transport and potentially leakage of intracellular components [52]. Valkova et al. (2001) compared the effect of methyl-, ethyl-, and propylparaben against *Enterobacter cloacae* (*E. cloacae*). They showed that the antimicrobial activity increased with increasing chain length with propylparaben (2.2 mM) being more efficient against *E. cloacae* than methylparaben (2.6 mM). As an indicator, they used the optical density (OD): no increase in OD was observed in the presence of propylparaben, an increase of approximately 2 OD in the presence of ethylparaben, and an increase of more than 5 OD in the presence of methylparaben. They attributed the greater antimicrobial activity of propylparaben to its increased hydrophobicity, resulting in a greater interference with the phospholipid bilayer, disruption of membrane transport processes, and inducing leakage of intracellular components [52]. Parker (1969) also studied the effect of a mixture of methylparaben (0.04% w·v^−1^ (2.6 mM)) and propylparaben (0.02% w·v^−1^ (1.1 mM)) on *Bacillus subtilis* (spore). It was shown that they inhibited spore germination resulting in a reduction of the cell formation from 40% in the control to only 5% in the paraben-containing medium at 32 °C after 4 h [53]. Additionally, Eklund (1984) showed that increasing carbon chain lengths of parabens can increase the proton transport into the cell, thereby collapsing the proton motive force [54].

#### 2.1.5. Benefits of Combining Preservatives 

Preservatives can be combined to extend their spectrum of preservation through a synergistic effect. A synergistic effect is defined as an enhancement in activity when several APs are combined compared to their effect as preservatives used individually [12]. On a microbiological base, synergy can occur if preservatives target different cellular sites in a certain microorganism [55]. Additionally, synergy also occurs if a combination of preservatives increases their activity against a greater microorganism spectrum. A broader preservative action can be obtained if a preservative which shows only antibacterial activity is combined with a second preservative which shows good antifungal activity [35,55]. Gupta & Kaisheva (2003) described that a combination of benzyl alcohol and chlorobutanol as well as benzyl alcohol and methylparaben have a higher efficacy against fungi compared to singly used preservatives [5]. Another synergistic effect against fungi and bacteria was obtained from the combination of methylparaben and chlorobutanol. The strongest positive effect has been found for the combination of methylparaben and propylparaben [5]. In a different study, Gilliland et al. (1992) found that the combination of methylparaben and propylparaben showed bactericidal action, whereas when used as single preservatives, the action was only bacteriostatic [56]. Besides their synergistic effect, parabens are often used in combination because their water solubility decreases with increasing chain length. Thus, combining parabens enables an antimicrobial protection in hydrophilic and hydrophobic phases of an emulsion [12,14,35]. For instance, the combination of methyl- with propylparaben is used in the topical gel Regranex (Bristol-Myers Squibb Co., New York, NY, USA) or in Fosatur Nasal Spray (Therapicon SRL., Milano, Italy). Another study carried out by Yano et al. (2016) claimed that the combination of benzalkonium chloride and benzyl alcohol has a synergic effect hypothesizing that benzyl alcohol increases the membrane permeability for benzalkonium chloride, thereby accelerating its accumulation inside the cell [47]. Additionally, preservative combinations might also lead to a greater protein stability [57]. This is the case for the combination of *m*-cresol and phenol which is used in numerous insulin formulations, e.g., Biosulin N (MJ Biopharm Pvt. Ltd., Mumbai, India); Insulatard Innolet (Novo Nordisk A/S., Bagsvaerd, Denmark). 

However, observed synergistic effects can sometimes be a result of an overall increased concentration of compounds. Thus, conclusions about synergistic effects must be investigated precisely and should be drawn carefully [12,35]. 

#### 2.1.6. Effects of Excipients on the Antimicrobial Activity of Preservatives

The antimicrobial efficacy of an AP can be increased in the presence of excipients which also have a destabilizing effect on microorganisms. The combination of such is defined as potentiation [12]. A characteristic example for this effect, is ethylenediaminetetraacetic acid (EDTA). As a chelating agent, it acts on the cell wall of microorganisms by chelating magnesium ions, leading to the release of lipopolysaccharide (LPS) from the outer membrane [58,59]. Therefore, the effect of many preservatives, including benzalkonium chloride, methylparaben, propylparaben, and phenol, can be enhanced in the presence of EDTA [12]. Interestingly, low concentrations of nonionic surfactants, e.g., polysorbates, can increase the efficacy of an AP as well. The improvement of the efficacy of the preservative in the presence of surfactants is attributed to the surfactant’s ability to permeabilize the bacterial cell to the AP. Although this effect is present, it only occurs over a small concentration range of surfactant. In the presence of polysorbates above their critical micelle concentration, preservatives can partition in the micellar system, thus reducing the concentration of preservative in the liquid phase [12,60,61,62]. For example, Fukahori et al. (1958) determined the concentration of parabens in aqueous and micellar phases [60]. They proceeded as follows. Methyl-, ethyl-, propyl-, and butylparaben were added to formulations including polysorbate 80 ranging from 0–8 µg·mL^−1^. For determining the concentration of aqueous paraben, the solutions were filtered with an ultrafiltration cell and ultrafiltration membrane at constant pressure (3 kg·cm^−2^) at 25 °C. Spectrophotometric analysis was used to determine the concentration of parabens in solutions without polysorbate 80. High-performance liquid chromatography (HPLC) was performed, as polysorbate 80 shows UV absorption properties, to determine the amount of preservatives in the paraben-polysorbate 80 system. Parabens found in the filtrate were regarded as free parabens in solution. As parabens have hydrophobic properties (alkyl and phenyl group) and hydrophilic properties (carbonyl group, phenolic hydroxyl group), they can interact with hydrophobic and hydrophilic regions of the nonionic surfactant. The partition coefficient (*K*) in the micellar phase increased with the increasing carbon chain length of the alkyl residue, e.g., *K* was greater than 2 for butylparaben but significantly below 1 for methylparaben. Additionally, the partition in the micellar phase was dependent on the hydrocarbon chain length of the surfactants. Increasing the surfactant chain lengths leads to greater incorporation of parabens into micelles. On the contrary, the paraben distribution in micelles decreased with the increasing polyoxyethylene chain length of the surfactant [60]. In general, incorporation into micelles can decrease the antimicrobial activity of a preservative as they are only active if they are free in aqueous solution and not incorporated within micellar structures [12,35].

### 2.2. Antimicrobial Efficacy Tests

Multi-dose formulations have to pass specific antimicrobial efficacy tests in order to guarantee the formulation’s ability to inhibit the growth of a broad range of microorganisms including Gram-positive and Gram-negative bacteria, yeasts, and molds [10]. Regarding the effect of APs, they are subdivided into “microbiostatic” agents which prevent microbial growth, e.g., by inhibiting microbial reproduction, and “microbicidal” agents which act irreversibly by killing microorganisms [13]. However, the addition of the preservative should not replace good manufacturing practices and aseptic production should still be ensured [8,10]. An important criterion of a multi-dose formulation is the designated period of utilization as the drug product must be protected from microbial contamination throughout the whole administration period. Hereafter, the criteria as described in the Ph. Eur. [10], the United States Pharmacopoeia and National Formulary (USP-NF) [8], and the Japanese Pharmacopoeia (JP) [9] are summarized. Additionally, a detailed comparison between the antimicrobial efficacy tests and the preparation of the test microorganisms was published by Moser & Meyer [63].

#### 2.2.1. Test for Efficacy of Antimicrobial Preservatives by Ph.Eur.

In the Ph.Eur., it is stated that the antimicrobial efficiency test shall be, if possible, carried out in the final container. The preservative system is regarded as suitable if there is a significant reduction or no increase in the number of microorganisms under the given conditions: At time zero, 10^5^–10^6^ CFU (colony forming unit)·mL^−1^ microorganisms are added to the container. Challenging microorganisms are chosen and listed in the pharmacopoeia based on their likelihood to contaminate a drug product. Ph.Eur. recommends the following microorganisms: *P. aeruginosa* (Gram-negative), *S. aureus* (Gram-positive), *C. albicans* (yeast), and *A. brasiliensis* (molds). For oral preparations, *E. coli* (Gram-negative) might be the bacterium of choice and *Zygosaccharomyces rouxii* (fungi) for drugs intended for oral preparations formulated with high sugar concentrations. The added volume of the microbial suspension should not exceed 1% of the product volume. For inoculation, the container must be kept at 20–25 °C protected from light [10]. After defined time points, samples are taken to determine the amount of living organisms. Methods recommended in the pharmacopoeia for counting the microorganisms are membrane filtration and/or plate count method. Afterwards, the log_10_ reduction over time is calculated. Depending on the administration route, the acceptance criteria differ: Parenteral formulations, eye preparations, intrauterine preparations and intramammary preparations are classified as category 1. Category 2 includes ear preparations, nasal preparations, preparations for cutaneous application, and preparation for inhalation. Last, category 3 includes oral preparations, oromucosal preparations and rectal preparations. For adequate protection, the Ph.Eur. differentiates between two criteria: Criterion A stands for the recommended efficiency that should be achieved by the AP. However, criterion A cannot be reached in some cases, e.g., if an increased AP concentration harmed the patient (safety issues). In such cases, criterion B must be achieved [10]. 

In Table 3, A and B criteria are listed according to the administration route.

#### 2.2.2. Antimicrobial Effectiveness Testing by USP

According to USP43-NF38 (General Chapter 51), any preservative added must be declared on the label. Microbial testing should be performed considering the nature of the product and its potential to cause adverse effects. The test shall be carried out in original containers if “a sufficient volume of product is available in each container and if the container can be entered aseptically, e.g., needle and syringe through an elastomeric rubber stopper” [8]. If this is not the case, the test can also be performed in sterile, capped containers being inert relative to the AP. The latter can be chosen if the microorganism cannot be transferred into the final container. Depending on the administration route, the antimicrobial efficacy must pass specific thresholds. It is differentiated between four categories: category 1 includes injections, other parenterals, e.g., otic products, emulsions, sterile nasal products, and ophthalmic formulations; category 2 includes topically used products made with aqueous bases or vehicles; oral products other than antacids are summarized in category 3; and antacids in category 4. According to the USP, the following microorganisms should be used for all four categories: *P. aeruginosa, S. aureus*, *C. albicans*, *A. brasiliens*, and *E. coli*. The added volume of microorganisms should be between 0.5% and 1.0% of the volume product. Regarding categories 1–3, the added concentration of microorganisms is between 1 × 10^5^ and 1 × 10^6^ CFU·mL^−1^ of the product. For antacids, the initial micro-organism concentration is 1 × 10^3^ – 1 × 10^4^ CFU·mL^−1^. After incubation, the log_10_ reduction of the initial microorganism concentration is determined after certain intervals via the plate-count procedure [8]. The criteria for each category are listed in Table 4. 

#### 2.2.3. Preservatives-Effectiveness Tests by JP

The preservatives-effectiveness tests of the JP are harmonized with the tests issued by the Ph.Eur. and USP. The effectiveness is determined by directly inoculating and mixing microbial test strains into the final product and subsequently assessing the amount of survived test strains over the test period. The testing environment should be sterile. Suitable aerobic microorganisms for the test are *S. aureus*, *B. subtilis,* and *P. aeruginosa* [9]. As anaerobic bacteria, the JP recommends using *Clostridium sporogenes* and as fungi *C. albicans* and *A. brasiliensis* [9]. The test is performed via membrane filtration or by direct inoculation of the microorganisms with the final product. Additionally, a negative control not including any microorganisms must be prepared for comparison. After having defined time points, the formulations are tested for microbial growth at 20–25 °C. The formulation shows adequate protection if no evidence for microbial growth is found within the test period. If there is evidence of microbial reproduction, the test criteria are not fulfilled, unless it can be clearly argued that the failure was caused by parameters unrelated to the examined product [9]. For instance, an invalid cause would be if microbial growth was also observed in the negative control. However, if the test is declared to be negative, it must be repeated using the same number of microorganisms as in the first attempt. If it meets the criteria the second time, the product is declared as sterile [9].

### 2.3. Physico-Chemical Properties of Antimicrobial Preservatives

#### 2.3.1. Concentration Range of Antimicrobial Preservatives

The efficacy of a preservative depends on its free concentration within the formulation. This means that an AP shows the highest activity if it is free in solution and not bound to other formulation components or adsorbed by packaging material [12,34]. For example, Heljo et al. (2015) reported that the ranking order with which different preservatives bound to a model protein (*m*-cresol > phenol > benzyl alcohol) correlated with the reduction in their antimicrobial efficacy [19]. Incompatibilities with packaging material are described in Section 2.5. When determining the appropriate concentration, two aspects are important: 

(i) The preservative’s concentration must be high enough to prevent microbial growth. For this, the MIC can serve as a guideline. According to the European Committee for Antimicrobial Susceptibility Testing (EUCAST), MICs are defined as “the lowest concentration of an antimicrobial agent that inhibits the growth of a microorganism” [64,65]. They are determined by counting the visible growth of microorganisms in microtitration plate wells filled with incremental dilutions of a preservative [64,65]. Observed differences between antimicrobial agents are supposed to be related to different metabolisms and inherent resistance against the preservative. Additionally, preservatives might have different target sites on or in the microorganism which can also lead to a difference in the MIC values between preservatives [36]. Examples of MICs determined in previous studies are listed in Table 5. Of all preservatives, benzalkonium chloride is by far the most efficient against microbial growth with MIC values ranging from 1.25 to 30 μg∙mL^−1^ (3.5 × 10^−3^–8.3 × 10^−2^ mM) [14].

(ii) The concentration should be kept to a minimum to prevent side reactions in the patient such as bronchoconstriction in the case of benzalkonium chloride [66,67,68] and endocrine disrupting effects in the case of parabens [69,70,71], or interactions with formulation components. 

Table 5 lists the most frequently used AP concentrations [3,12,26] in biological drugs as well as their MIC values against different microorganism classes. Microorganism classes cover Gram-negative bacteria (*P. aeruginosa*, *E. coli*), Gram-positive bacteria (*S. aureus)*, yeasts (*C. albicans)*, and molds (*A. niger*).

**Table 5 pharmaceutics-15-00563-t005:** Concentration range of antimicrobial preservatives in protein formulations and the MIC (minimum inhibitory concentration) values measured for microorganisms [14,17,72,73]).

Preservative	Typically Used AP Conc. /% w∙v^−1^	MIC/μg∙mL^−1^	
Benzalkonium chloride	0.3–0.6	*P. aeruginosa*	8.3 × 10^−2^
		*S. aureus*	3.5 × 10^−3^
		*E. coli*	4.4 × 10^−2^
Benzyl alcohol	83.2–101.7	*P. aeruginosa*	18.5
		*S. aureus*	0.1
		*C. albicans*	23.1
		*A. niger*	46.2
		*E. coli*	1.5
Chlorobutanol	up to 28.2	*Gram-positive*	3.7
		*Gram-negative*	5.6
		*Yeast*	14.1
		*Fungi*	28.2
*m*-Cresol	13.9–27.7	*P. aeruginosa*	n/a
		*S. aureus*	n/a
		*A. niger*	n/a
		*C. albicans*	n/a
Methylparaben	5.9–11.8	*P. aeruginosa*	26.3
		*S. aureus*	13.1
		*C. albicans*	13.1
		*A. niger*	6.6
		*E. coli*	6.6
Phenol	95.6–141.3	*P. aeruginosa*	19.1
		*S. aureus*	10.6
		*A. niger*	3.3
		*E. coli*	26.7
Phenoxyethanol	36.2–72.4	*P. aeruginosa*	23.2
		*S. aureus*	61.5
		*C. albicans*	39.1
		*A. niger*	23.9
		*E. coli*	26.1
Propylparaben	0.6–1.1	*P. aeruginosa*	>5.5
		*S. aureus*	2.8
		*C. albicans*	1.4
		*A. niger (ATCC 9642)*	2.8
		*E. coli (ATCC 9637)*	2.8

#### 2.3.2. pH

When choosing a pH value for a multidose biological formulation, the preservative’s activity as well as the API’s stability must be ensured [12]. In general, the efficiency of an antimicrobial preservative depends on the pH as either the AP itself is altered or the interactions with the microorganism are changed [34]. Positively charged antimicrobial agents will interact most strongly with the negatively charged outermost microbiological shell at elevated pH values, whereas the antimicrobial efficiency of agents, that work on intracellular components, is increased in their un-ionized form compared to their ionized form [12,13]. For example, phenol and *m*-cresol show the highest antimicrobial activity at pH values below 9 [12,14]. Phenoxyethanol shows a broad antimicrobial activity over a wide range from pH 3–10 [13,14,74].

Benzyl alcohol is most efficient at pH below 5 and has only little antimicrobial effects at values above pH 8 [14]. For example, Karabit et al. (1986) investigated the pH-dependence of benzyl alcohol regarding its antimicrobial activity. In particular, they calculated decimal reduction time values (D-values) which represented the microorganisms’ resistance against the benzyl alcohol present at 1%. The pH-profile showed a maximum at pH 6.0 (log(D) > 1.5) and a minimum between pH 4 and 5 (log(D) < 0.5). Considering a pH range of 3–8, the D-values slightly decreased with decreasing pH values, indicating an increased preservative activity in acidic conditions. [48]. 

Methyl- and propylparaben require pH values below 8 as the formation of the phenolate anion at higher pH values decreases the overall antimicrobial activity. However, they still have a broad antimicrobial efficiency range from pH 4 to 8 [12,14]. In their study, Aalto et al. (1953) described how methyl-, ethyl-, propyl-, and butylparaben successfully inhibited the growth of *A. niger* over a pH range of 4–7. A slightly reduced activity was observed for propyl- and butylparaben at pH 8. The reduced activity was explained by the fact that at higher pH values, the pK value of the acid is reached. As a consequence, the phenolate anion was formed, resulting in a reduced activity [75]. 

Chlorobutanol needs the most acidic conditions as it shows considerably reduced activity at pH values greater than 5.5 [14]. Patwa & Huyck (1966) reported that chlorobutanol was stable between pH 3 and 6 in buffered solutions and between pH 3 and 4 in unbuffered solutions. During storage at room temperature, the preservative was reported to have 90% stability for 45 days when stored at pH 5–6 [76]. 

Benzalkonium chloride has an overall broad antimicrobial spectrum ranging from pH 4–10 [12,14,48]. As shown by Karabit et al. (1988), the antimicrobial activity increased with increasing pH values due to an increased surface attraction. They calculated decimal reduction time values (D-values) which represent the microorganisms’ resistance against the benzalkonium chloride present at 0.001%. D-values decreased with increasing pH indicating a reduced microbial resistance at higher pH values. For example, the log(D) decreased from about 15 at pH 3.0 to about 5 at pH 8.0 [48]. In addition, the pH has also an impact on the charge of the microorganism’s cell surface. With increasing pH, the surface gets more negatively charged. Therefore, positively charged preservatives, e.g., quaternary ammonium compounds, such as benzalkonium chloride, are more attracted to the surface and consequently, have a higher efficiency [34].

Moreover, the interactions between APs and proteins are affected by the pH value [20,23]. Evers et al. (2019) reported a decreased peptide solubility at pH 4.5 and 5.2 when phenol or *m*-cresol were added to the formulation compared to peptide alone. With increasing pH (6.5 and 7.4), no solubility issues were observed [20]. On the contrary, Thirumangalathu et al. (2018) showed that a pH reduction from 7.0 to 3.5 decreased the aggregation rate of recombinant human granulocyte colony-stimulating factor (rhGCSF) caused by benzyl alcohol. At acidic pH, rhGCSF had a highly positively charged surface what increased protein-protein electrostatic repulsion. As a result, rhGCSF stayed stable even when benzyl alcohol was added. In contrast, protein aggregation was accelerated at pH 7.0 by the presence of benzyl alcohol. At pH values > 7.0, the protein was neutrally charged which increased intermolecular protein-protein interactions. They hypothesized that these favorable interactions also facilitate benzyl alcohol-induced structural perturbation. Furthermore, they suggested that the absence of aggregation at lower pH value was due to the increased colloidal stability of the protein [23]. 

As a summary, the preservatives pH ranges as well as the pH values found in licensed protein drugs are depicted in Table 6.

#### 2.3.3. Solubility

As APs differ in their hydrophilicity, solubility is another parameter that must be considered when developing a multi-dose formulation. Especially parabens are difficult to solubilize in water as they have very hydrophobic properties (methylparaben: 1 g in 400 mL water and propylparaben: 1 g in 2500 mL water at 20 °C) [14,24]. Chlorobutanol might also be difficult to solubilize [14]. For each AP, the solubility in water and 95% ethanol is listed in Table 7.

#### 2.3.4. Toxicity 

The addition of preservatives to a medicinal formulation requires special justification and should be avoided if possible [77]. With regard to toxicological side effects induced by preservatives, the European Medicines Agency (EMA) guidelines on excipients in the dossier for application for Marketing Authorization of a Medicinal Product (EMEA/CHMP/QWP/396951/2006) states “Antimicrobial preservatives are normally added to prevent microbial proliferation arising under in use conditions. These properties are due to certain chemical groups which are usually harmful to living cells and might therefore be associated with certain risks when used in humans. Thus, inclusion of antimicrobial preservatives or antioxidants in a medicinal product needs special justification. Wherever possible the use of these substances should be avoided, particularly in case of paediatric formulations. The concentration used should be at the lowest feasible level.” In general, adverse reactions caused by APs are less likely since they are used at very small concentrations. However, some were suspected to cause side effects, or even death, in several studies and case reports [66,67,68,71,78,79,80,81,82,83,84,85]. Selected examples of case studies and toxicology studies are discussed below.

##### Benzalkonium Chloride

It was reported that benzalkonium chloride-containing aerosolized asthma medicate caused dose-dependent bronchoconstriction and collapse of the respiratory tract [66,67,68,70]. In a case report, a 17-year-old female asthmatic patient showed no bronchodilator response (multi-dose nebulizer containing 50 µg benzalkonium chloride per dose of albuterol 2.5 mg). On the contrary, the patient was in a severe health condition with symptoms, e.g., respiratory distress, unable to speak, and increased temperature. Throughout the treatment, the patient received accumulatively around 32 mg of benzalkonium chloride mixed in albuterol solution over 3.5 days. After treating the patient with a benzalkonium chloride-free albuterol solution, a rapid improvement of her respiratory status was observed [67]. Other case reports and studies showed similar dose dependent bronchoconstriction effect with increasing benzalkonium chloride concentration [66,68]. In 2017, the European Medicines Agency (EMA) published a report about the use of benzalkonium chloride in medicinal products for human use. They issued that all benzalkonium chloride-containing drugs intended for inhalation must include the following information in the package leaflet “Benzalkonium chloride cause wheezing and breathing difficulties (bronchospasm), especially if you have asthma”. Additionally, benzalkonium chloride was attributed to cause swelling inside the nose when used in nasal formulations [70]. However, even though several case studies reported a correlation between long exposures to benzalkonium chloride containing drugs and adverse effects, it was not possible to determine a general threshold for the use of benzalkonium chloride in medicinal products for human use [70]. According to the EMA, data evaluating the effect of benzalkonium chloride containing drugs on children is rare or missing; and if available, no differences between children and adults were found. As most data were gained from toxicity studies in animals, the need for extrapolation makes it impossible to establish a clinical ‘safe threshold’ [70]. 

##### *m*-Cresol

Over the last few years, adverse effects caused by *m*-cresol have been reported and studied by several research groups. For the oral route, a LD_50_ of 242 mg·kg^−1^ bodyweight was calculated for an undiluted single application, and for the dermal route, the LD_50_ was calculated to be 2050 mg·kg^−1^ bodyweight [86]. *m*-Cresol is often present in rapid-acting insulin formulations. However, insulin administration via continuous subcutaneous insulin infusion was reported to increase infection at the infusion site possibly due to the presence of *m*-cresol in such formulations [87]. A study carried out by Faassen et al. (1989) correlated the infection rate with the type of preservative included in the insulin formulation: 54% of patients (13 of 24) using *m*-cresol insulin formulations reported inflamed infusion sites, whereas only 19% (5 of 26) reported inflamed infusion sites using methylparaben-containing insulin [87]. In a follow-up study, Faassen et al. (1990) evaluated the effect of preservative containing insulin formulations on the function of leucocytes. The function was tested by determining the ability of leucocytes to kill *S. aureus* after 60 min of incubation. The presence of *m*-cresol and methylparaben decreased the leucocyte function to 74.8% and 85.0%, respectively, whereas the insulin formulations without any preservative had a killing percentage of 95.4%. No killing of bacteria was observed in control mixtures without leucocytes. Thus, the results of this findings showed a negative influence of preservative on the efficacy of leucocytes to kill *S. aureus* which might potentially increase the risk of infection at the infusion site [88]. In a different study, Paiva et al. (2016) analyzed the biophysical interactions between *m*-cresol and the lipid bilayer in membrane models focusing on the effect of *m*-cresol at the cellular-membrane level [89]. The background of their study was to elucidate cytotoxic effects of *m*-cresol, as this excipient is used in pharmaceutical formulations of insulins and vaccines up to concentrations of 30 millimolar. They used different molar dilutions of *m*-cresol ranging from 1:1 to 1:10^4^ and the medicinal product Humulin (Eli Lilly and Company, Indianapolis, IN, USA) which contains human insulin at 100 U·mL^−1^ and *m*-cresol at 2.5 mg·mL^−1^. The lipid molecules contained among other lipids *N*-palmitoyl-sphingomyelin (PSM) which is a sphingomyelin found in higher portions in animal cell membranes and cholesterol (Chol) which is the central sterol in animal cell membranes. Interactions of pure *m*-cresol and Humulin formulation with model bio-membranes were studied via intrinsic fluorescence properties and atomic force microscopy (AFM) in the presence and absence of a lipid ordered and liquid disordered phase as well as the combination of both. Both analysis methods confirmed that *m*-cresol interacts with model-membranes, and that it preferably interacted with more ordered membrane domains, e.g., cholesterol-sphingomyelin enriched domains. For instance, it was shown by Paiva et al. (2016) via steady-state fluorescence anisotropy that the preservative’s mobility decreased in the presence of liposomes compared to aqueous solution. Furthermore, in-situ real-time AFM was performed to analyze the effect of *m*-cresol on morphological and structural changes of the lipid bilayer [89]. The results showed that interactions occurred mainly at the interface of ordered and disordered domains leading to reduced area of PSM/Chol-enriched domains over time, and even elimination of these domains at high *m*-cresol concentrations (300 µM) after 2 h. Hence, it was concluded that the preservative-induced structural perturbation was both concentration- and time dependent. Last, it was shown that *m*-cresol induced transversal cell leakage in three different neuron types, including a neuroblastoma cell line, neurons from the peripheral nervous system, and neuronal cells from the central nervous system. By using a cell voltage clamp approach, it was shown that the presence of *m*-cresol led to a concentration-dependent increase of leak conductance [89]. The leak current in neuroblastoma cells increased from less than 25% (3 µM *m*-cresol) to ca. 125% (300 µM *m*-cresol). Taken the results together, the findings of Paiva et al. (2016) are of medicinal interest as there are many formulations including *m*-cresol on the market. Once *m*-cresol is administered intravenously, it can get in contact with cell membranes, and might interact with them as described above [89].

##### Parabens

In the past, parabens were reported to cause endocrine-disrupting effects [69,83]. The endocrine system consisting of internal organs is an important contributor for the maintenance of the homeostasis in the human body. Its function can be affected by internal and external factors, leading to endocrine disrupting effects [83]. As described by the WHO, an endocrine disruptor “is an exogenous substance or mixture that alters function(s) of the endocrine system and consequently causes adverse health effects in an intact organism, or its progeny, or (sub)populations” [90]. A detailed review about parabens was published by Nowak et al. (2018) focusing on the use of parabens, their metabolism, and toxicity [83]. Regarding the parabens’ impact on endocrine organs, Vo et al. (2010) reported that parabens increased the weight of the adrenal glands and decreased the weight of the thyroid gland in model rats suggesting a correlation between the presence of parabens and an imbalanced hormone system [85]. Additionally, parabens were reported to affect the steroid sex hormones of male and female animals and humans. For instance, Oishi (2002) described how male rats produced less testosterone when exposed to propylparaben. Propylparaben was administered at doses of 0, 0.01, 0.1, and 1.00%. The serum testosterone concentration was reduced in a dose dependent manner being the highest (64.6%) in the group that was fed with the highest propylparaben concentration (1.00%) compared to the control group. The testosterone content was reduced to 90.3% in the group exposed to 0.01% propylparaben [71]. Furthermore, several groups confirmed that parabens disturbed the concentrations of female steroid sex hormones. Aker et al. (2016) reported a dosage-dependent increase in the concentration of sex-hormone-binding globulin in pregnant women after exposure to methylparaben. Additionally, methylparaben and propylparaben were attributed to a decrease in the concentration of estradiol [84]. 

A study carried out by L. Barr et al. (2012) investigated the relationship between paraben levels in human breast tissue collected from 40 breast cancer patients [91]. Their studies revealed that, among 160 human breast tissue samples, 158 showed quantifiable levels of at least one paraben with the highest mean levels for *n*-propylparaben (16.8 ng per gram tissue (range from 0–2052.7 ng)) and methylparaben (16.6 ng per gram tissue (range from 0–5102.9 ng)). However, no statistically significant correlation between the paraben concentration and tumor location could be found. Additionally, no correlation was found between the paraben level and the content of estrogen receptors [91]. 

Because of these claims, the committee for medicinal products for human use (CHMP) evaluated the toxicity of parabens, stating that the “use of methylparaben up to 0.2% of the product is not a concern for humans including the pediatric population whatever the age group” [69]. With regard to propylparaben, a NOEL of 100 mg·kg^−1^·day^−1^ was determined. Thus, a Permitted Daily Exposure (PDE) of 2 mg·kg^−1^·day^−1^ of propylparaben was calculated for use in pediatric patients and adults. Taking the toxicity reports of parabens into consideration, they decided that additional information on the package leaflet of medicinal products for human were not required as no sufficient, clinical-relevant evidence regarding the impact of parabens on human health was present [69].

##### Benzyl Alcohol

A detailed substance evaluation conclusion document about the toxicity of benzyl alcohol was published in 2021 by the Federal Institute for Occupational Safety and Health, Germany (EC No 202-859-9, 2021) [92]. Another very detailed review about the toxicology and dermatology of benzyl alcohol was published by Scognamiglio et al. (2012) [93]. As benzyl alcohol was reported to cause side effects when used in pharmaceutical products [78,79,80,81,94], the EMA published a report about the use of benzyl alcohol as an excipient [78]. After uptake, it is rapidly metabolized to benzaldehyde in the liver by alcohol dehydrogenase. Later on, benzaldehyde is oxidized to benzoic acid by aldehyde dehydrogenase. In the last step, benzoic acid is eliminated by its transformation to hippuric acid which is then secreted as part of the urine [78]. Thus, mutations in the genes encoding for enzymes involved in the metabolism of benzyl alcohol may interrupt its normal degradation and may lead to toxicity. Additionally, immaturity of the detoxification pathway can lead to toxic side reactions in neonates [78]. Indeed, Lebel et al. (1988) reported that neonates who were treated with benzyl alcohol-containing phenobarbital were reported to have increased benzyl alcohol level as their metabolism is not fully developed yet. Higher levels of benzoic acid were found in the serum of preterm neonates compared to term neonates. A higher benzyl alcohol concentration was found in their urine as benzoic acid, and the level of hippuric acid (normal degradation product of benzyl alcohol in the liver via an oxidation process) was reduced compared to term neonates [79]. In the 1980s, various reports were published discussing the use of benzyl alcohol as related to the cause of sixteen neonatal deaths at two medicinal centers [80]. In the report by Gershanik et al. (1982), five pre-term neonates were treated frequently with injections of heparinized sodium chloride solution containing 0.9% benzyl alcohol as “flushes” for intravenous lines [95]. They were treated with other drugs which were reconstituted in benzyl alcohol-containing water. The estimated dose of benzyl alcohol was 99–234 mg·kg^−1^·day^−1^. All dead neonates showed an increased level of benzyl alcohol (50 to more than 200 mg·mL^−1^) and of hippuric acid in the urine. When the use of benzyl alcohol was stopped, no other cases were observed. In the other report, the death of 10 neonates was attributed to benzyl alcohol poisoning [96]. All the neonates had at least one central venous or arterial catheter flushed with saline solution including 0.9% benzyl alcohol. The daily amount of benzyl alcohol was calculated to be approximately 191 mg·kg^−1^·day^−1^. Increased serum benzoic levels were reported being 8.4 to 28.7 mM (normal value 0 mM). After stopping the treatment with benzyl alcohol, the serum concentration of benzoic acid decreased from 14.4 mM to 0 mM. The syndrome described in these fatal benzyl alcohol poisoning is referred to as “gasping syndrome” [78,80,81]. It is characterized by metabolic acidosis and respiratory depression [80,81]. Reynolds (1990) suggested a correlation between a hand-held albuterol nebulizer containing benzyl alcohol in its bronchodilator diluent and significant hemoptysis. In a study, a nonsmoking patient had three episodes of significant hemoptysis after reconstituting the albuterol solution in benzyl alcohol containing reconstitution medium. When changing to a preservative-free albuterol formulation, the hemoptysis was not observed. Thus, Reynolds (1990) advised not to include benzyl alcohol in formulation intended for inhalation [97]. 

In view of all these aspects, the EMA considered benzyl alcohol as an excipient with known side effects. Toxicity problems were attributed to benzyl alcohol accumulation and consequent metabolic acidosis in the human body. In addition to age, the ethnic polymorphism of alcohol dehydrogenase may also influence toxicity. However, it is recommended that benzyl alcohol not be used in medicines for children up to three years of age [78]. 

##### Phenoxyethanol

Phenoxyethanol can be regarded as safe when at a concentration up to 1% w·v^−1^ (72.4 mM) [14,98]. Therefore, adverse reactions induced by phenoxyethanol are less likely as the most commonly used concentration of phenoxyethanol in protein formulations is between 0.5% and 1.0%. Harmful reactions were only seen in animal studies when the exposure was significantly higher than patients would be exposed to. The Scientific Committee on Consumer Safety (SCCS) (2016) declared phenoxyethanol as safe at a maximum concentration of 1.0% in their assessment, including children of all ages. They conducted an acute oral toxicity study in rats reporting a LD_50_ in 1840 mg·kg^−1^ and 4070 mg·kg^−1^ for females and males, respectively. In an acute dermal toxicity study in rabbits, no mortalities occurred, extrapolating the LD_50_ to > 2214 mg·kg^−1^ bodyweight [98]. 

##### Summary

Based on toxicology studies of the different preservatives, the European Chemical Agency set the following LD_50_ (Table 8). 

#### 2.3.5. Allergenicity

Certain of the presented preservatives have been described in relation to allergenicity, e.g., immediate hypersensitivity reactions, such as benzalkonium chloride, benzyl alcohol, or *m*-cresol [99,100]. These reactions were reported using different types of medications. *m*-Cresol immediate hypersensitivity was reported for insulin type medications, benzalkonium chloride for ophthalmic products and benzyl alcohol with parenteral medications [101,102,103] (see Caballero and Quirce (2020) and references cited therein [99]). A recent review paper by Díaz et al. (2021) summarizes the current knowledge on drug allergy by also considering the used excipients [102]. Anaphylactic reactions were reported for vitamin B12 injections containing benzyl alcohol [104,105]. Common adverse reactions, such as urticaria angioedema, asthma, and lung irritation, were reported for the preservatives benzoates and parabens [106].

### 2.4. Interactions of Preservatives with Excipients

The activity of the preservative should not be influenced by the presence of excipients. Incompatibilities with excipients, e.g., insoluble ingredients, nonionic surfactants, or sequestering agents can impact the preservative’s activity as well as physico-chemical properties of the solution [12]. For instance, the antimicrobial activity of parabens, phenolic derivatives, and alcohols was reported to be considerably reduced in the presence of nonionic surfactants, e.g., polysorbate 20 (PS20) or polysorbate 80 (PS80), as hydrophobic preservatives were prone to be incorporated into the surfactants’ micelles [12,14,62].

In semi-solid preparations (pastes, gels, creams and ointments) or suspensions, it is possible that the AP binds to solids resulting in a decrease of its activity [12,35]. Batuyios & Brecht (1957) described that quaternary ammonium compounds, such as benzalkonium chloride, were bound by kaolin and talc [107]. Additionally, since all APs have a hydrophobic structure, they are prone to be incorporated into micelles by emulsifier or detergents [12]. With its doughnut-like structure, cyclodextrin incorporates APs into its hydrophobic core reducing the free AP concentration [12]. Chin et al. (2010) described that parabens can be adsorbed by β-cyclodextrin linked by hexamethylene diisocyanate from aqueous solution. The adsorption capacity was calculated to be 0.0305, 0.0376, 0.1854, and 0.3026 mmol·g^−1^ for methyl-, ethyl-, propyl- and butylparaben, respectively, indicating that the amount of adsorption is directly correlated with the alkyl chain length [108]. Loftsson et al. (1992) investigated the interactions between chlorobutanol, benzalkonium chloride, methylparaben, and propylparaben with 2-hydroxypropyl-β-cyclodextrin. They reported a decreased antimicrobial activity of chlorobutanol, methylparaben, and propylparaben due to complex formation. On the contrary, benzalkonium chloride, which is very hydrophilic, was still active against a wide range of microorganisms. Thus, they concluded that the efficacy of an AP depends on their ability to form complexes with 2-hydroxypropyl-β-cyclodextrin, with more hydrophobic APs showing a greater efficacy reduction [109]. 

As mentioned above, nonionic micellar surfactant systems (e.g., PS20 and PS80) decrease the antimicrobial efficacy [61,62,110]. For example, Barr & Tice (1957) studied the antimicrobial activity of 48 agents, including methylparaben (0.18% w·v^−1^ (11.8 mM)), propylparaben (0.15% w·v^−1^ (8.3 mM)), phenol (0.5% w·v^−1^ (53.1 mM)), cresol (0.5% w·v^−1^ (46.2 mM)), chlorobutanol (0.5% w·v^−1^ (28.3 mM)), and benzalkonium chloride (0.1% w·v^−1^ (2.8 mM)), in the presence of 5% w·v^−1^ PS20. 10 mL of preservative-containing solution was mixed with 0.1 mL of either *P. aeruginosa* for 24 h or *Monilia albicans* and *A. niger* for 72 h. Of the 48 preservatives studied, benzalkonium chloride was one of only eight antimicrobial agents able to inhibit the growth of microbes. For all other preservatives, microbial growth was observed, attributed to the reduced antimicrobial activity due to complex formation between phenolic preservatives and the polyether structures of PS20 [62]. 

In another study, Blanchard et al. (1977) postulated that interactions between phenolic preservatives and nonionic surfactants include two binding sites in the surfactant micelles. One was attributed to have a high affinity but a low capacity for preservatives, whereas the other exhibited a negligible affinity but almost unlimited capacity for preservatives. The high affinity/low capacity binding site was proposed to be located near the hydrocarbon core (=hydrophobic environment) whereas the low affinity/high capacity was proposed to be found in the polyoxyethylene (POE) region of the micelle (hydrophilic environment). It was hypothesized that hydrophobic preservatives migrated from the thermodynamically less favorable aqueous phase into the thermodynamically favored hydrophobic core of polysorbate micelles. It was further reported that migration into the micellar core was more pronounced for preservatives with higher hydrophobicity. If the capacity in the high affinity region were saturated, the preservative would start partitioning in the POE region. Even though this phase is more hydrophilic, it is thermodynamically favored compared to the even more hydrophilic aqueous phase [111].

Even though these incompatibility issues are known, many protein formulations including preservatives and polysorbates do exist on the market. Recently, studies were conducted focusing on interactions between preservatives and polysorbates [112,113]. These studies differed from earlier study as they did not analyze the impact of polysorbates on the antimicrobial activity of preservatives but focused on the impact of preservatives on the stability of polysorbates [114]. For example, Shi et al. (2015) studied whether *m*-cresol is compatible with trace levels of PS20 and PS80 focusing on whether the known incompatibility issues are dependent on the polysorbate concentration [113]. To study the polysorbate-*m*-cresol interactions, they used HPLC-CAD (charged aerosol detector). The authors also developed a highly sensitive method for PS20 and PS80 quantitation with a limit of quantification (LOQ) of 5 ppm. They analyzed the compatibility in water and with a peptide (ca. 6 kDa) formulated in phosphate buffer at pH 7.0. No information about the peptide concentration was provided. First, they added 2.8 mg·mL^−1^ (25.9 mM) of *m*-cresol into 20 mL of PS20 or PS80 solutions ranging from 10 ppm to 100 ppm, and to one solution containing 800 ppm PS80. Solutions including 100 ppm PS20 or PS80 turned slightly turbid in the presence of *m*-cresol, an effect which was even greater at 800 ppm PS80. The visual inspection confirmed the incompatibility problem indicating that the incompatibilities increase with increasing polysorbate concentration. Moreover, the polysorbate concentration was determined at room temperature, protected from light, after 28 days. In the presence of *m*-cresol, the concentration of both polysorbates decreased significantly at 50 ppm with a relative PS20 loss of 14% and a relative PS80 loss of 20%. This effect was even greater at 100 ppm being 26% in the case of PS20 and 34% in the case of PS80. At lower PS levels, e.g., 10 ppm and 20 ppm, no obvious reduction in the polysorbate concentrations were observed. Furthermore, PS20 and PS80 were spiked to the peptide formulation at 20 ppm PS20 (0.002%) and 50 ppm PS80 (0.005%). In conclusion, Shi et al. (2015) confirmed that higher levels of polysorbate 20 and 80 (50 ppm, 100 ppm, 800 ppm) are incompatible with *m*-cresol leading to increased turbidity of the solution and loss of polysorbate content due to PS degradation. In comparison, PS80 was less compatible with *m*-cresol than PS20. However, their study lacked the discussion of the underlying mechanism of preservative-PS interactions [113]. The findings of Shi et al. (2015) could be confirmed by Gilbert et al. (2020), reporting that the addition of *m*-cresol (3.15 mg·mL^−1^ (29.1 mM)) to a PS80 solution (1 mg·mL^−1^) increased solution turbidity and irreversibly PS80 micelle morphology changes. They used small-angle neutron scattering (SANS) which is a useful analysis method for determining the structure, size distribution of micelles, and their aggregate formation. The evaluation is based on intensity measurements of scattered neutrons giving insight into the aggregation number, aggregate surface area, and radius of gyration, and time-dependent changes in micelle composition. Gilbert et al. (2020) described that the scattering intensity was increased in the presence of *m*-cresol compared to solutions containing only PS80 at 22 °C suggesting the existence of larger PS80 micelles in the solutions with *m*-cresol. Additionally, the presence of *m*-cresol induced micellar growth, as indicated by an increased scattering intensity after three days [112]. Very interestingly, it was hypothesized by invariant analysis of the scattering data that the total mass of micellar aggregates did not increase over time, suggesting that no new aggregates were formed and no additional *m*-cresol or PS80 were incorporated into the micelles. Thus, increasing micellar aggregate sizes may be caused by fusion or coalescence. The authors showed that the micellar aggregate formation is temperature-dependent being reduced but not fully prevented at 4 °C (scattering intensity ~ 10^2^ cm^−1^ after 3 days) compared to 37 °C (scattering intensity > 10^3^ cm^−1^ after 3 days). Additionally, it was shown that the micelle aggregation depended on the buffer condition. PS80/*m*-cresol solution turned immediately turbid after mixing with citrate buffer, whereas the solution stayed clear for several hours when diluted in water. The authors divided *m*-cresol-induced micellar aggregate formation into two phases. The first step includes the rapid micellar aggregate formation immediately after the preservative was added. In the second phase, micellar aggregates increase in size as a result of coalescence. In addition, the micellar aggregation rate is influenced by temperature and buffer composition [112].

In a further study, Torosantucci et al. (2018) analyzed the interactions between 0.3 mg·mL^−1^ (3.2 mM) phenol and 2 mg·mL^−1^ PS20 based on NMR analysis [21]. Preservative-surfactant interactions were determined by comparing NMR spectra of phenol depending on different PS20 concentration. No line broadening could be observed in the presence of any tested PS20 concentration, indicating that no phenol–PS20 interactions occurred [21]. In comparison, Shi et al. (2015) [113] and Gilbert et al. (2020) [112] reported preservative–PS interactions, but they also analyzed interactions with a different preservative, namely *m*-cresol. 

### 2.5. Incompatibilities with Packaging Material and Consumables

One major field in drug development is package development. Package development is of great importance to guarantee stability, safety, and efficacy of the drug product. In general, packaging material can be divided into primary, secondary, and tertiary packaging. Primary packaging is the material which is in direct contact with the product, consisting of a container and a seal. One important characteristic of seals is that they do not interact with the product or the environment [115,116]. Incompatibilities of preservatives with packaging material, indicated by adsorption onto, diffusion into, and diffusion out of the material, were reported leading to reduced preservative concentrations in solution over time [117]. Consequently, the formulation might not show sufficient protection against microorganisms after prolonged storage [14,24,118,119]. In the following, observed incompatibilities with rubber stoppers, plastic containers, and consumables, e.g., silicone tubes, are discussed.

#### 2.5.1. Incompatibilities with Rubber Stoppers

Rubber stoppers belong to the primary packaging material, thus are in direct contact to the drug product [115,116]. All herein discussed preservatives have hydrophobic properties thus have the potential to be incorporated into rubber stoppers. This issue was investigated by Royce & Sykes (1957) analyzing the time-dependent loss of phenol, *m*-cresol, benzyl alcohol and chlorobutanol from multi-dose containers closed by “red” or “white” rubber stoppers. No information about the differences and composition between red and white rubber, e.g., their material composition, and the exact concentration unit of the tested bacteriostats was provided in their paper. In brief, the authors showed losses of bacteriostats from rubber-closed multidose containers of injections. They calculated water -partition coefficients to predict the distribution of several preservatives in aqueous solutions and rubber closures (Table 9). To understand the losses of the tested bacteriostats and the underlying mechanism of these losses by absorption, a diffusion and volatilization process is discussed [118]. 

Several studies showed that Tear-off seals consisting of three aluminum layers fastened at the upper side of rubber stoppers could delay the loss of preservatives [120,121]. For instance, Landi et al. (1968) monitored the phenol concentration in different primary packaging materials [121]. One outcome was that the phenol content (0.3%) of a tuberculin PPD (Purified Protein Derivative) solution did not change during two years of storage at 5 °C, whereas phenol loss was observed for stoppered glass vials. Their glass vials were closed by either pre-treated or untreated “yellow” and “white” rubber (rubber composition not further specified). Pre-treated rubber stopper were incubated at 25 °C in 0.3% phenol. Optionally, rubber stoppers were sealed by three-piece aluminum caps to analyze the preservative loss by evaporation. Preservative loss was unaffected by the type of rubber stoppers but affected by the temperature and vial size (for more details, see Landi et al. 1968 [121]). Pre-treating the rubbers reduced the percentage of lost phenol, depending on the used stopper. The evaporation rate of phenol was tested by adding three-piece aluminum caps in three different ways including (a) partially sealed (only the lower part of three-piece aluminum caps was applied), (b) completely sealed (complete aluminum cap was applied), and (c) completely coated seal (sealed vial was dipped into melted paraffin to coat the complete seal and part of the vial neck). In summary, these tests showed that preservative losses due to evaporation can be reduced by proper (full) sealing, e.g., using three-piece aluminum caps [121]. Additionally, Held & Landi (1985) compared the loss of phenol closed by “yellow” rubber and either tear-off seals composed of three-piece aluminum layers or flip-off seals composed of a plastic top connected to two aluminum layers. For comparison, the rubber stoppers were sealed partially (only lower part of three-piece aluminum seal was applied) or completely (complete tear- off or flip-off seal). A solution of 0.3% phenol was stored in large glass vials (8.6 mL capacity) and small glass vials (2.3 mL capacity) at 37 °C in an upward and horizontal position for 36 months. The preservative loss was independent of the horizontal or upward position but differed greatly between the vial size and the sealing. For example, the half-life (50% loss) of phenol was estimated to be 8.4, 2.8, and 2.0 years for the tear-off seal, the flip-off seal, and the partially sealed vials, respectively being greatly reduced to 1.4, 0.63 and 0.46 years in the smaller vials, respectively. When stored at 5 °C, the preservative loss was significantly reduced relative to 37 °C, e.g., 58.9 years for the large vials sealed by tear-off caps. No loss could be observed for any vial size and sealing option when stored at −28 °C for 21 months. Furthermore, it was described that flip-off seals were not as airtight as tear-off seals, e.g., the air flow through an unsealed rubber was 42 mL·min^−1^ being only minimally reduced to 41.2 mL·min^−1^ if sealed by flip-off seals, but greatly reduced to 3.7 mL·min^−1^ if sealed by tear-off seals [120].

To sum up, the work by Royces & Siekes (1957), Landi et al. (1968), and Held & Landi (1985) showed that preservative concentration can be reduced in solution depending on the rubber type potentially minimizing the antimicrobial efficacy of the formulation. However, it was also demonstrated that the concentration reduction can be minimized by sealing caps. It was shown that tear-off seals are more sufficient in preventing preservative loss compared to flip-off seals [118,120,121]. Nowadays, coated stoppers are available that might reduce the issues described above.

#### 2.5.2. Incompatibilities with Plastic Container

Over the last decades, more plastic containers are used as primary packaging material as they are cheaper, shatterproof, and lighter compared to glass container [122]. However, incompatibilities with plastic containers were also observed for several preservatives [119,122,123]. Roberts et al. (1979) recognized that preservatives with increased affinity to polyethylene, represented by an increased hexane-water partition coefficient, showed a greater reduction [119]. They analyzed the kinetics of benzyl alcohol loss from aqueous solutions into polyethylene containers with a wall thickness of 0.71 mm. The containers were composed of three parts including the actual container, a dropper nozzle, and a screw cap. As a negative control, the preservatives were also stored in a sealed glass ampoule under the same conditions. It was reported that for preservatives with a hexane-water partition coefficients smaller than 1, as is the case for benzyl alcohol (0.17), the kinetics followed a mono-exponential reduction, whereas the loss of preservatives occurred with a hexane-water coefficient greater than 1, e.g., chlorocresol with 2.2, followed a bi-exponential kinetic [119]. 

Moreover, the group of Amin analyzed the compatibility of several preservatives (commonly used in ophthalmic preparation) after storage in low density polyethylene (LDPE) blow-fill-seal packs and polypropylene (PoP) blow-fill-seal packs at 40 °C/25% relative humidity [122,123]. Despite the cost saving advantages, blow fill seal tanks have a high sorption potential, thus leading to incompatibilities with hydrophobic excipients. They reported that the loss of preservatives from LDPE packs followed the order chlorobutanol (=chlorbutol) (96%) > benzyl alcohol (72%) > propylparaben (50%) > methylparaben (12%) > benzalkonium chloride (approximately 3%) with lipophilic volatile preservatives (chlorobutanol and benzyl alcohol) showing greater losses compared to lipophilic nonvolatile preservatives (methyl- and propylparaben). The losses were attributed to permeations (chlorobutanol and benzyl alcohol) and sorption (parabens), whereas the loss of benzalkonium chloride was regarded as negligible. It was thought that the resistance of benzalkonium chloride against absorption was due to its ionized form. The authors suggested that losses of chlorobutanol and benzyl alcohol happened in two steps starting with the preservatives’ partition in the polymer matrix followed by an irreversible desorption to the atmosphere. These findings highlighted that volatility might also play a major role in the mechanism of preservative loss. A preservative’s loss can also be impacted by formulation variables like pH, buffer strength, and concentration. They also stressed that environmental factors such as humidity could affect the sorption rate of parabens. However, it should also be mentioned that different preservative concentrations were used in the experiments (0.015% w·v^−1^ for parabens (methylparaben (1.5 mM), propylparaben (0.8 mM)), 0.5% w·v^−1^ for alcohols (benzyl alcohol (46.2 mM), chlorobutanol (28.3 mM)), and 0.01% w·v^−1^ for benzalkonium chloride (0.3 mM)) which could also have had an effect on the preservative reduction. To sum up, all tested preservatives were compatible with polypropylene blow-fill-seal packs, whereas only benzalkonium chloride was compatible with LDPE packs [122].

#### 2.5.3. Incompatibilities with Consumables (Flexible Tubes)

Preservatives can already be lost during manufacturing steps. Several studies focused on the loss of preservatives due to flexible tubing. For example, Saller et al. (2017) analyzed the loss of benzyl alcohol (10 mg·mL^−1^ (92.5 mM)), phenol (5.5 mg·mL^−1^ (58.4 mM)), and *m*-cresol (3.15 mg·mL^−1^ (29.15 mM)) in placebo formulations (no protein) from silicone tubing during filling processes [117]. They analyzed different tubing types: two platinum-cured silicone tubing types with inner diameters (ID) of 6.0 mm and 1.6 mm and a wall thickness of 2.1 mm and 1.6 mm, respectively, and Fluidvit FPM tubing composed of fluoropolymer with an ID of 6.4 mm and a hardness of 60 Shores. Based on the absorption maximum λ_max_ of the preservatives (benzyl alcohol: λ_max_ = 259 nm, phenol: λ_max_ = 270 nm, and *m*-cresol: λ_max_ = 272 nm), UV measurements in the range of 220–350 nm were performed to determine the concentration of the preservatives. In general, the loss of preservatives depended on the type of tubing. The two silicone tubing types caused a rapid preservative loss whereas no preservative losses were detected for any of the three preservatives in the fluoropolymer tubing type; additionally, the loss in the silicone tubing types was approximately five times greater for the smaller inner diameter compared to larger. For example, the loss of phenol and benzyl alcohol was determined to be 3%, and the loss of *m*-cresol 10% in silicone tubing with an ID of 6.00 mm after 5 min of incubation. Incubation in tubes with an ID of 1.6 mm induced a significantly higher preservative loss of 20–40%. After 6 h, the preservatives were completely lost in 1.6 mm ID silicone tubing. In 6.0 mm ID tubing, 63% of benzyl alcohol, 53% of phenol, and 34% of *m*-cresol remained in solution. Based on the experimental outcomes, Saller et al. (2017) concluded that the most decisive parameter for influencing the total loss was the partition coefficient *k*, with higher values indicating a greater solubility in the silicone matrix. The following *k*-values were determined: *k* (*m*-cresol): 0.83 > *k* (benzyl alcohol): 0.44 > *k* (phenol): 0.27. Table 10 summarizes the beforementioned results.

The importance of *k* became even more clear when the flux *J* of the three preservatives through the silicone rubber (ID 6.0 mm) was calculated based on Fick’s equation: *m*-cresol had the greatest flux values (e.g., 0.5 g·m^−2^·s^−1^) compared to benzyl alcohol (0.26 g·m^−2^·s^−1^) and phenol (0.21 g·m^−2^·s^−1^). It was reported that the diffusivity *D* was strongly influenced by the molecular weights of the preservatives with smaller weights leading to greater *D* values. For instance, benzyl alcohol and *m*-cresol, which both have a molecular weight of 108.14 g·mol^−1^, showed similar diffusivity values of 2.3 × 10^−10^ m^2^·s^−1^ and 2.2 × 10^−10^ m^2^·s^−1^, respectively, whereas the diffusivity of the smaller phenol (94.11 g·mol^−1^) was increased by approximately 30% being 3.0 × 10^−10^ m^2^·s^−1^. Finally, they analyzed the evaporation rate from the tubing surface by measuring the weight of tubing pieces after incubation with preservatives. The calculated evaporation half-life time correlated with the vapor pressures found in the literature [124,125,126] as the lowest half-life time was found for phenol which also had the highest vapor pressure (0.35 mmHg at 25 °C [126]). In detail, the evaporation rate was the highest for phenol (186 min), followed by benzyl alcohol (296 min) and *m*-cresol (377 min). To sum up, Saller et al. (2017) showed, for the test tubing items, that losses were most pronounced with *m*-cresol, followed by phenol and benzyl alcohol. As shown in this study, the selection of an appropriate tubing type, e.g., fluoropolymer tubing, can reduce the amount of lost preservative [117]. These findings were in accordance with Bahal & Romansky (2001), who reported that the concentration of methylparaben (0.18% w·v^−1^ (11.8 mM)) and propylparaben (0.02% w·v^−1^ (1.1 mM)) did not decrease in fluoropolymer-based tubing like Teflon or Vitube at room temperature (22–25 °C) after 6 h. When incubated in silicone tubing, e.g., Silastic, approximately 20% of methylparaben and almost 70% of propylparaben were lost at room temperature after 6 h. Thus, greater sorption was reported for the more hydrophobic preservative. Temperature and pH had no impact on the amount of sorbed preservative. It was also noticed that the amount of sorbed preservative increased with increasing surface area of silicone tubing, e.g., a surface area of 20 square inches caused a decrease of propylparaben by approximately 20% at pH 3.5, whereas a surface area of 80 square inches led to a decrease of approximately 50% [127]. These findings are in contradiction to those of Saller et al. (2017), who reported that a smaller inner diameter of the tubing led to an increased loss of preservative [117]. In a follow-up study, Bahal & Romansky (2002) also included benzyl alcohol and benzalkonium chloride into their tests regarding the effect flexible tubing on the preservative concentration. The results showed that all preservatives apart from benzalkonium chloride were absorbed by silicone tubing types. The concentration of benzalkonium chloride decreased by only 3% after incubation with Silastic silicone tubing at 25 °C after 48 h. They interpreted this result as “essentially no sorption” hypothesizing that benzalkonium chloride’s resistance to sorption was attributed to its ionic nature. However, they found out that benzyl alcohol was lost to ~30% after six hours of incubation due to sorption by silicone tubing types, and almost 50% were lost after 120 h at 25 °C. No sorption occurred in solutions incubated with Teflon, Zelite, and FEP laminated Tygon [128]. In accordance with Saller et al. (2017), the sorption rate was independent of the initial concentration [117]. Thus, it was hypothesized that the sorption process of preservatives was rather diffusion-controlled than binding/adsorption-controlled. Pre-soaking of silicone tubing type (Amesil) with benzyl alcohol for 168 h showed a decreased sorption rate of only 19% after 168 h of incubation [128].

In conclusion, the studies by Bahal & Romansky (2001, 2002) [127,128] and Saller et al. (2017) [117] showed that the concentration of hydrophobic preservatives in solution can be reduced due to absorption by silicone tubes and subsequent diffusion into the atmosphere. Thus, fluoropolymer-based tubes should be used for filling processes to prevent preservative loss during manufacturing.

#### 2.5.4. Summary

In Section 2.5.1, Section 2.5.2 and Section 2.5.3, incompatibility issues with packaging materials and consumables are discussed. Table 11 summarizes selected incompatibility issues of preservatives with packaging material, consumables, as well as recommended containers if mentioned in the reference [12,14,24].

### 2.6. Stability and the Fate of Storage Conditions of Antimicrobial Preservatives

In multi-dose formulations, preservatives should be stable during storage time and over the intended application period. This section provides selected information about the stability and storage conditions of the preservatives covered in this review.

#### 2.6.1. Benzalkonium Chloride

The chemical stability of benzalkonium chloride may be impacted by metals, air, and light [14]. In a study carried out by Assem & Fouda (2019), it was shown that benzalkonium chloride is adsorbed onto the surface of carbon steel obeying Langmuir adsorption isotherm. Based on a ΔG_ads_ of less than −20 kJ·mol^−1^, the adsorption mechanism on carbon steel in 1 M HCl was attributed to electrostatic attraction between the positively charged benzalkonium chloride and the charged metal surface (physisorption). Based on a calculated negative ΔH_ads_, the authors concluded that the adsorption of the cationic benzalkonium chloride is a spontaneous, exothermic process [130].

Bin et al. (1999) [131] investigated the mechanism(s) of adsorption of benzalkonium chloride to filter membranes and analyzed the effect of formulation and processing parameters on the adsorption of benzalkonium chloride. They analyzed six different sterilizing grade filter membranes of the following composition and properties: (i) polyvinylidene fluoride (PVDF) membrane modified with a poly(acrylate)ester to obtain hydrophilic membrane properties, (ii) PVDF membrane modified with a poly(acrylate) ester and a poly(cationic amine) exhibiting a net positive charge, (iii) PVDF membrane modified with a nonionic agent for hydrophilic membrane properties, (iv) Nylon 66 (hydrophobic polymer) cast on a polyester support, (v) Nylon 66 membrane modified to reduce the adsorption of proteins, and (vi) polyethersulfone (PES) membrane modified with poly(acrylate)ester, with a surface coating being impermeable. Their results showed that the adsorption properties of each filter membrane depended on its chemical composition. The reduced absorption of benzalkonium chloride was observed for membranes being hydrophilic and nonionic or having hydrophilic and cationic properties. Hydrophobic or anionic membranes exhibited a significant benzalkonium chloride adsorption. In addition, they studied specific formulation parameters, including the concentration of benzalkonium chloride, the presence of a tonicity modifying agent (sodium chloride, mannitol, glycerin), and the presence of a chelating agent (edetate disodium). The greatest increase in preservative adsorption occurred when sodium chloride or edetate disodium was used with membranes that contained cationic sites. For these membranes, a benzalkonium chloride reduction up to 60% was observed. The processing parameters investigated by the authors were: flow rate, temperature range from 25 to 37 °C, autoclaving (121 °C for 20 min), interrupting the flow (to mimic shutdown of the processing line up to 1 h), and pre-saturating the filter membrane. Autoclaving had essentially no effect on any of the membranes studied for the absorption experiments. The effect of flow rate was small up to be negligible. The rate of adsorption was inversely related to flow rate and the extent of adsorption was inversely related to temperature. Pre-saturating the filter membranes was investigated as a means of preventing adsorption of benzalkonium chloride during filtration. Bin et al. (1999) proposed that a pre-saturation of the filter membrane with benzalkonium chloride was an effective method to reduce the adsorption of benzalkonium chloride during filtration. From a mechanistic point of view, benzalkonium chloride-filter membrane adsorptions followed the Langmuir equation [131].

Furthermore, Amin et al. (2012) studied the loss of benzalkonium chloride from low density polyethylene (LDPE) and polypropylene (PoP) blow-fill-seal packs (BFS) which are often used for ophthalmic formulations. It was shown that the amount of lost preservative was independent of the humidity as no significant differences were observed when stored at 25% relative humidity and 75% relative humidity [122]. After storage at 40 °C/25% relative humidity for 90 days, the preservative concentration was reduced by 3.43% in the glass control, 2.7% in LDPE-BFS, and 1.9% in PoP-BFS. The preservative losses were determined to be negligible indicating that benzalkonium chloride could be used in LDPE-PFS packs and PoP-BFS packs. However, adsorption to polyethylene container can be considerably increased if counteracting ions are present in the solution. For example, the percentage of adsorbed benzalkonium chloride onto polyethylene container was increased by more than 20% in the presence of 0.05% ammonium thiocyanate [129]. This example shows that preservative interactions with packaging material can be greatly influenced by the presence of counterions.

#### 2.6.2. Benzyl Alcohol

Benzyl alcohol slowly oxidizes to benzaldehyde and subsequently to benzoic acid when exposed to air. It is incompatible with strong acids and oxidizing agents [14,24]. It may be stored in metal or glass container [14,118]. Plastic containers should not be used except for polypropylene containers or vessels coated with inert fluorinated polymers like Teflon. For instance, an aqueous solution of 2% v·v^−1^ benzyl alcohol stored in a polyethylene container showed a loss of 15% of the original benzyl alcohol content after 13 weeks of storage at 20 °C [119]. In a different study by Amin et al. (2012), it was shown that benzyl alcohol loss increased by approximately 10 times to almost 80% in low density polyethylene containers after storage at 40 °C/25% humidity for 90 days compared to less than 10% in polypropylene container [122]. Additionally, benzyl alcohol is slowly adsorbed by natural rubber, neoprene, and butyl rubber closures. For example, Royce & Sykes (1957) showed that of a 1% benzyl alcohol solution stored in a 2 mL cartridge closed by an untreated rubber, only 0.9% of benzyl alcohol was detected in the solution after one month of storage. No losses were observed when the rubber was first equilibrated in a benzyl alcohol-containing solution prior to the storage, indicating that pre-soaking in a preservative solution leads to saturation in the rubber, thereby reducing the loss of preservative [118].

#### 2.6.3. Chlorobutanol

Chlorobutanol is a volatile preservative with a vapor pressure of approximately 1.6 × 10^3^ Pa at 20 °C [132]. Its volatility might be a critical aspect when choosing the right packaging material as a correlation between its increased volatility and increased preservative loss from plastic containers and closures was reported in previous studies [118,122]. It is only stable at acidic pH values as degradation in aqueous solution is catalyzed by hydroxide ions. For example, Nair & Lach (1959) calculated the half-life of chlorobutanol at 25 °C being 90 years at pH 3 but only 3 months at pH 7.5 [133].

Studies by Lachman et al. (1963) showed that chlorobutanol is absorbed by neoprene rubber. They stored the vials at 60 °C in an upright and inverted position. The loss was determined by chemical (UV absorbance) and microbiological methods (antimicrobial efficacy of chlorobutanol). The chemical analysis showed that after 30 days, 59% of the chlorobutanol was still present when stored upright and 53% when stored upside down. Interestingly, the percent of remaining preservative analyzed by the microbiological method was higher compared to the chemical method being ca. 70% in the upright and inverted position after 30 days of storage. In their study, the author discussed the observed effects in the context of a “contribution of rubber extractives and preservative degradation products towards enhanced antimicrobial activity” [134]. In a different study, they were able to identify the preservative-induced extractables by their UV absorbance [135]. Different rubber types were analyzed, including neoprene rubber, natural rubber, and butyl rubber. For the neoprene rubber closure, the primary accelerator (imidazoline type) and the secondary accelerator (thiazole type) could be identified as extractives. For the natural rubber, the identified extractables were its primary accelerator (thiazole type) and an antioxidant (substituted butyl phenol). Last, the extractables of butyl rubber could be identified as its primary accelerator (substituted carbamic acid) and its secondary accelerator (thiazole type) [135]. Thus, the apparently higher chlorobutanol concentration detected by the microbiological assay relative to the chemical method could be attributed to the presence of extractables which contributed to the increased antimicrobial activity of the solution [134,135]. Furthermore, Friesen & Plein (1971) reported that chlorobutanol was incompatible with polyethylene containers, resulting in a decreased antimicrobial efficacy compared to a glass control. The loss was shown to be temperature dependent. Stored in the refrigerator for ten weeks, 95% of chlorobutanol were measured in a 30 mL polyethylene container whereas only 90% remained in solution when stored at room temperature. An even greater decrease was found when stored at 45 °C [136]. Polyethylene incompatibility was confirmed by Amin et al. (2012) who reported a chlorobutanol loss of almost 100% after storage in low density polyethylene containers at 40 °C/25% humidity for 90 days, whereas only approximately 20% were lost in polypropylene containers [122]. No correlation was found between the percentage of lost preservative and increasing the concentration from 0.5 mM to 2.0 mM [122].

#### 2.6.4. *m*-Cresol

*m*-Cresol is not suspected to hydrolyze under environmental conditions [137]. With a Henry’s law constant of 0.087 Pa·m^3^·mol^−1^ at 25 °C, the preservative has only a low likelihood volatility from aqueous solutions [137]. Additionally, Seraghni et al. (2012) reported that *m*-cresol was photodegraded after irradiation at 365 nm: in a single system containing only the preservative, the photodegradation efficacy was 30% after 6 h of irradiation, whereas in a *m*-cresol/Fe(III)-citrate system, the degradation efficacy increased to 52% upon 6 h of irradiation [138]. The higher degradation could be correlated to the formation of hydroxyl radicals which, as mentioned before, led to photodegradation of *m*-cresol. It was also shown that the photo-degradation was pH-dependent being 71% at pH 2.86 and 54% at pH 6.44 after 6 h of irradiation; low pH value also resulted in a higher reduction rate of Fe(III) to Fe(II) which consequently increased the amount of hydroxyl radicals. They also demonstrated that the presence of oxygen has an important role in the photochemical reaction as it can cause the formation of radical species in aqueous solutions, thus increasing the photodegradation rate of *m*-cresol [138]. Furthermore, *m*-cresol can darken in color over time and when exposed to light and air [14].

#### 2.6.5. Parabens

Aqueous solutions of parabens are stable at pH 3–6 for up to four years when stored at room temperature. However, they are prone to hydrolyzation at pH 8, e.g., the percentage of hydrolyzed propylparaben is approximately 6%. Increasing resistance of hydrolysis is shown with the increasing alkyl chain length of the parabens [75].

Gmurek et al. (2015) studied the photodegradation of single parabens as well as mixtures thereof. The purpose of the study was on photochemical degradation of hazardous water contaminants, namely methyl-, ethyl-, propyl-, butyl-, benzylparaben, and p-hydroxybenzoic acid (individually or in mixture) using ultraviolet C lamps in the presence and absence of hydrogen peroxide. In short, the authors showed a total removal of individual pollutant being achieved after 2 h or 6 min for UVC and UVC/H_2_O_2_, respectively. This study illustrates that photodegradation needs to be considered [139].

Moreover, Kakemi et al. (1971) studied preservative-plastic interactions by heating the solution at 100 °C for 30 min. Polyvinyl chloride containers containing increased amounts of plasticizers showed the highest adsorption rate for all parabens, with 42.2 µg uptake per gram plastics for methylparaben, 125.2 µg·g^−1^ for ethylparaben, 323 µg·g^−1^ for propylparaben, and 456 µg·g^−1^ for butylparaben. In polyethylene container, the adsorption rate increased with increasing carbon chain length from methylparaben (7.6 µg·g^−1^) to butylparaben (68.5 µg·g^−1^). The same ranking order was found for polypropylene ranging from 0 µg·g^−1^ (ethylparaben) to 28.8 µg·g^−1^ (butylparaben). As the same ranking order was shown for different plastic containers, it was suggested that the adsorption rate increased with the increasing carbon chain length of the parabens [14,129].

In addition, Amin et al. (2012) investigated the sorption behavior of methyl- and propylparaben in blow-fill seal packs commonly used for ophthalmic solutions [123]. The main components of blow-fill seal packs are polyolefins, including polyethylene (most commonly low-density polyethylene (LDPE)) and polypropylene (PoP). Despite the cost saving advantages, blow fill seal tanks have a high sorption potential, thus leading to incompatibilities with hydrophobic excipients. As parabens are often used in ophthalmic preparations, the target of this study was to analyze the interactions between the preservatives and LDPE and PoP blow fill seal (BFS) tanks depending on the packaging material (LDPE or PoP at 40 °C/25% relative humidity), pH (pH values: 3, 5, 7 at 40 °C/25% relative humidity), paraben concentration (0.5, 1.0, and 2.0% w·v^−1^ at 40 °C/25% relative humidity), buffer species (acetate and phosphate at 40 °C/25% relative humidity), and relative humidity (40 °C/25% relative humidity and 40 °C/75% relative humidity). The paraben concentration (0.015% w·v^−1^ for methyl- and propylparaben if not stated otherwise) was determined via HPLC system. Studying the effect of the blow-fill seal tank material, it was reported that both parabens showed a significantly higher loss at all three pH values when stored in LDPE packs compared to polypropylene packs. The reduction in the concentration of methylparaben was approximately 10× in LDPE compared to PoP packs, whereas the loss of propylparaben was even more increased being approximately 25× higher in LDPE bags compared to PoP backs. After three months of storage, methylparaben loss was determined to be between 10% and 12% and propylparaben loss to be between 37% and 42% when stored in LDPE packs. Only small amounts (not more than 4%) were lost in PoP packs after three months. As PoP contains an additional methyl group which is not found in LDPE, the authors suggested that the higher diffusivity rate of parabens in LDPE packs is caused by its greater chain mobility, and thus reduced barrier properties. They also attributed the higher propylparaben loss to its greater log*P* value (propylparaben: log*P* = 3.04; methylparaben: log*P* = 1.96) since it is more hydrophobic than methylparaben. Analyzing the effect of pH did not show any significant differences in the sorption rate at different pH values. Moreover, 0.01 M acetate buffer at pH 5, and 0.01 M phosphate buffer at pH 3 and 7 were compared to an unbuffered methylparaben solution as the control. No differences were observed for acetate buffer at pH 5 and phosphate buffer at pH 3. However, a significant reduction in preservative loss from approximately 15% to 10% was observed for the phosphate buffer at pH 7 from LDPE packs after 90 days of storage. In addition, increasing the buffer strength to 0.05 M showed an increased sorption suppression of methylparaben from 15% to 9% in LDPE and from 2.2% to 1.3% in polypropylene backs at pH 7, an effect that was only observable at 40 °C/25% relative humidity but not at 40 °C/75% relative humidity. Furthermore, increasing the concentration of parabens also increased the amount of sorption. The fraction of adsorbed preservative to increased concentration decreased over time, indicating that the system might reach saturation. Finally, they reported that the preservative loss was increased at 75% relative humidity compared to 25% relative humidity correlating this effect to an altered chain relaxation and in the packaging material at different humidity levels. In conclusion, Amin et al. (2012) showed that the loss of parabens is increased in low density polyethylene blow-fill-seal packs compared to polypropylene, and that the loss can be impacted by formulation variables, e.g., pH, buffer strength, and concentration. They also stressed that environmental factors such as humidity could affect the sorption rate of parabens [123].

#### 2.6.6. Phenol

Unlike alcohols, phenol is a weak acid. Phenol is stable in aqueous solutions, but can be oxidized by oxygen, especially by reactive oxygen species (Devlin, H., 1984) [140]. Phenol is very quickly photodegraded in air by OH radicals. Exposed to light and air, it can change the color to reddish or brownish with the coloring effect being affected by metallic impurities and oxidizing agents [14]. For example, in a study by Prasse et al. (2018) [141], phenol was oxidized by OH radicals produced from an UV/H_2_O_2_ process. The authors reported that approximately 2% of the lost phenol was transformed into α, β-unsaturated dialdehyde. Small amounts of 2-butene-1,4-dial were formed during the direct UV photolysis of phenol. No phenol degradation or 2-butene-1,4-dial formation was found when the preservative was exposed to light with a wavelength above 290 nm. Additionally, Royce & Sykes (1957) described that phenol interacted with a polyethylene container, leading to a reduced preservative concentration from 1% to 0.80% in solutions depending on the used stopper. Thus, glass, or polypropylene containers are recommended [14,24,118].

#### 2.6.7. Phenoxyethanol

Hydrolysis of phenoxyethanol was studied at pH 4.0, 7.0, and 9.0 at 50 °C. Less than 10% of the initial preservative concentration were hydrolytically degraded. Thus, phenoxyethanol can be considered as hydrolytically stable in aqueous solutions with a half-life of more than one year (echa.europa.eu). The photodegradation in air depends on the sunlight intensity and was extrapolated to be 50% after 11.8 h using an OH radical as a sensitizer at a concentration of 500,000 molecules·cm^−3^. The photo transformation rate in water was calculated to be max. 0.0045 per day with an overall elimination of 0.3%. As the extrapolated half-life time was estimated to be 5120 days, the likelihood of direct photolysis can be considered as negligible in aqueous solutions (ECHA, n.d.). Its activity as an antimicrobial agent might be reduced in polyvinyl chloride containers due to absorption [14]. A reduced phenoxyethanol content was reported by Lee (1984) when stored in polyvinyl chloride containers. In glass containers, no reduction in polyethylene content was observed [142].

### 2.7. Antimicrobial Preservatives Depending on the Administration Route

Based on multi-dose protein formulations already on the market, APs can be categorized regarding uses for different administration routes. For parenteral administration, *m*-cresol, phenol, and benzyl alcohol are mostly used [3,12,15]. Benzalkonium chloride plays a major role in nasal protein formulations, followed by chlorobutanol and a mixture of methylparaben and propylparaben. Furthermore, benzalkonium chloride is the preservative of choice for ophthalmic preparations. No AP-containing protein formulation has been found for inhalation. So far, only small molecule formulations for inhalation including an AP exist. Figure 1 shows the number and percentage distribution of preservative-containing market products in relation to the application route, including products containing one preservative as well as preservative combinations and products with antibacterial water. Figure 2 shows in more detail the relationship between the administration and the respective preservative.

Table 12 lists the concentrations of preservatives used in market products containing proteins or peptides as API depending on the application route.

### 2.8. Limitations for the Use of Antimicrobial Preservatives

Antimicrobial preservatives are usually active against any living cells, and thus can be harmful to humans, too. Therefore, preservatives should only be added to a formulation if absolutely necessary. For instance, important criteria are the likelihood of microbial contamination of the drug product or the maximum period of use after the first administration. There are some restrictions for the use of preservatives: According to the Ph.Eur. Monograph 0520 “Parenteral preparations”, antimicrobial preservatives can be added to injections at appropriate concentrations. No antimicrobial preservatives are added if the formulation is intended for single dose application exceeding 15 mL or if the administration route, for medical reasons, does not allow the use of preservatives, e.g., epidural, intrathecal, intracisternal, intra- or retro-ocular administration, or any route giving access to the cerebrospinal fluid [143]. Additionally, preservatives are not included in infusions (not to be confused with injections) and freeze-dried products [143,144].

## 3. Interactions between Antimicrobial Preservatives and Biological Compounds

Interactions between preservatives and biological compounds were studied in the past by several research groups. Preservatives were tested regarding their effect on the stability and aggregation tendency of the biological compound. It was reported that preservatives may decrease the stability of the tested biological compound mainly due to aggregation [5,15,16,17,18,19,20,21,22,23,145]. Two studies were conducted analyzing the effect of different preservatives on palmitoylated peptides [19,20]. However, as in the case of R6-Zn-insulin hexamer, there are also examples where preservatives actually increase protein stability [57]. Whittingham et al. (1998) showed that the presence of phenolic derivates, such as phenol and *m*-cresol, which were originally added to preserve the formulations, induced the formation of more stable zinc hexamers. These observations showed that the use of preservatives is not only limited to their function as antimicrobial agents, but they can be used as stabilizers as well [57]. This section focuses on factors affecting protein stability in biopharmaceuticals, the molecular interactions between proteins and preservatives, and how to counteract preservative-induced protein destabilization and aggregation. Additionally, we summarize selected studies regarding peptide-preservative interactions. Because the primary amino acid sequence of peptides is much shorter than that of proteins, they do not have the typical tertiary structure as found in proteins. Thus, interactions between peptides and preservatives might differ from those found with proteins [145].

### 3.1. Protein Stability in Biopharmaceuticals

When developing a multi-dose protein formulation, the protein itself is susceptible to degradation or protein particle formation (“aggregation”) induced by excipients, manufacturing processes, or storage conditions [146]. In general, protein instabilities can be distinguished between chemical and physical instability. Chemical instability includes reactions which lead to new chemical subunits by making or breaking covalent bonds, e.g., deamidation or oxidation. On the other side, physical instabilities do not change the chemical entities of a molecule but alter the physical state of it. Examples for physical instability are protein aggregation, surface adsorption, denaturation, or precipitation; the latter one can result from protein aggregate growth, but also from proteins being salted out, e.g., by an excluded solute [147,148]. As preservatives were reported to cause protein aggregation and denaturation, the correlated theoretical background is discussed in more detail.

Denaturation defines a process of which a protein has lost its globular, three-dimensional structure. As a result of unfolding and denaturation, the protein’s physical state is altered but its covalent bonds remain unchanged. Protein denaturation can be induced by chemical denaturants, e.g., guanidinium chloride, or temperature. A higher denaturation temperature is directly correlated with an increased conformational stability as more energy is required to unfold the protein. In most cases, thermal denaturation is irreversible because the rapidly unfolding protein molecules aggregate [147,149].

Much work has been done to understand protein aggregation in the last years. Aggregates can form based on non-covalent associations of polypeptide chains or on covalently bound peptide species. Very importantly, protein aggregation should be prevented as it can lead to severe side effects in the patient [147]. For instance, it can lead to immunogenic responses [150]. Increased antibody concentrations can also lead to problematic outcomes, e.g., the binding of neutralizing antibodies can affect the pharmacokinetic of the active pharmaceutical ingredient [151]. Moreover, aggregates may impact the bioavailability of the pharmaceutical [152].

In order to control protein destabilization induced by preservatives, it is important to focus on the protein’s intrinsic conformational stability as well as protein-protein interactions. Indeed, conformational stability is a very important factor as protein aggregates often arise from partially unfolded species. Protein stability can be increased by the addition of excipients. Roughly, they can be categorized into buffer, surfactants, sugars and other polyols, amino acids, amines, and salts [147,148]. For example, protein aggregation was shown to be reduced in preservative-containing formulations by the addition of sucrose and trehalose [16,18,23,153]. Sugars are preferentially excluded agents which minimize the amount of partially unfolded protein species as they are excluded from a protein’s first hydration shell. Consequently, the protein-water chemical potential is increased [153].

### 3.2. Studies about Protein—Preservative Interactions and Peptide-Preservative Interactions

A general overview regarding possible effects of protein/peptide on preservatives and vice versa is shown in (Table 13).

More details about published case studies concerning the interactions of preservatives with peptides, proteins, and antibodies summarized to elucidate current knowledge and approaches related to preservative interactions are found in the following sections.

#### 3.2.1. Peptide—Preservative Interactions

Heljo et al. (2015) studied the interactions of an acetylated model peptide with benzyl alcohol, *m*-cresol, and phenol focusing on peptide self-interactions and antimicrobial efficiency [19]. The acylated model peptide had a MW of approximately 4.5 kDa. It was added at 0, 1, 5 or 10 mg·mL^−1^. Benzyl alcohol was added at 8–12 mg·mL^−1^ (74.0–110.1 mM), phenol at 4–6 mg mL^−1^ (42.5–63.8 mM), and *m*-cresol at 1.5–3 mg mL^−1^ (13.9–27.7 mM). The antimicrobial efficiency of the preservatives in the presence and absence of the peptide was tested as stated in the Ph. Eur. A reduced activity was observed with increasing peptide concentrations ranging from 0 to 10 mg·mL^−1^. In many cases, the increase in peptide concentration lowered the formulation’s antimicrobial efficacy from the European Pharmacopoeia Standard A to B, or even to fail the test at all. In detail, increasing peptide concentrations led to the greatest reduction when combined with *m*-cresol followed by phenol; the least impact was observed for benzyl alcohol. Interestingly, the same ranking with which the preservative’s activity was reduced correlated with increased molecular size and self-interactions of the peptide. For example, the hydrodynamic radius measured by dynamic light scattering (DLS) and molar mass of peptide oligomers measured by composition-gradient multi-angle light scattering (CG-MALS) were significantly greater (from 2.5 nm to almost 4 nm, and from 25 kDa to more than 40 kDa, respectively) when *m*-cresol was added, but stayed almost constant in the presence of phenol or benzyl alcohol. The authors attributed the increased molecular size to the formation of reversible peptide oligomers which was accelerated by preservative-peptide interactions. To further analyze the underlying interactions, the adsorption of the preservatives onto peptide molecules was analyzed via nuclear magnetic resonance (NMR) in dependence of the peptide concentration. Based on the observed diffusion coefficient (*D_obs_*), the percentage of peptide-bound preservatives was determined. The results showed that the mobility of the preservatives added at 5 mg·mL^−1^ decreased with increasing protein concentration from 0 to 10 mg·mL^−1^. The ranking in which preservatives were bound onto the peptide followed the same order as the results obtained by light scattering analysis and antimicrobial efficiency (CR > PH > BA). Based on the correlation between peptide-bound preservative and reduced antimicrobial activity, the authors concluded that a preservative was the most efficient in its free form, and that binding to other components decreased its activity. In conclusion, Heljo et al. (2015) showed that preservative–protein interactions do not only increase peptide oligomerization, but also reduce the antimicrobial efficiency of the preservative. The ranking order in which interactions were observed correlated with the octanol:water partition coefficient with greater values leading to greater interactions (CR (1.98) > PH (1.48) > BA (1.05)). However, they also stated that the interactions with the peptide might not only depend on the hydrophobicity of a preservative, but also on hydrogen-bonding and π-interactions [19].

D’Addio et al. (2021) investigated the potential impact of phenol, *m*-cresol, and benzyl alcohol on the stability of a linear palmitoylated peptide [145]. The concentration of preservatives differed in the range of 0–113 mM *m*-cresol, 0–425 mM phenol, and 0–200 mM benzyl alcohol. Dynamic light scattering, nephelometry, SEC coupled with MALS, and DSC were performed to assess preservative-induced changes in the formulation and the kinetic of acyl-peptide. A fibrillation propensity with and without antimicrobial preservatives was monitored using the thioflavin T fibrillation screening assay. The DLS results measured 2-nm peptide species in formulations, including 2% benzyl alcohol, whereas 30-nm species were found in formulations, including 0.2% of phenol. These findings were in correlation with SEC-MALS results where only one peak with a molecular mass of approximately 23 kg·mol^−1^ (attributed to 5–6 mers) was found in formulations containing benzyl alcohol. On the contrary, several peaks were detected in formulations including phenol with peptide populations ranging from 20 to 2700 kg·mol^−1^. The turbidity of formulations including phenol decreased with increasing temperature from 14 NTU at 5 °C to 4 NTU at room temperature, indicating a dissociation of peptide species upon equilibration at room temperature. Increasing the temperature to 37 °C led to an even greater turbidity decrease to less than 1 NTU. No changes in the turbidity profile of the formulation including benzyl alcohol were observed even after five freeze-thaw cycles. Together with the DSC results, which showed an endothermic transition of the phenol containing formulation at 35 °C but no events in the benzyl alcohol containing formulation, it may be concluded that only phenol, but not benzyl alcohol, induced the formation of peptide species when stored in the refrigerator, but the formation was found to be reversible. Additionally, they looked for a blue shift in the tryptophan fluorescence emission which is only found if tryptophane is present in a more hydrophobic environment. As a decrease in the fluorescence of tryptophan is thought to correlate with conformational changes, fluorescence emission spectra give insight into excipient interactions, peptide self-association and aggregation. Such a blue shift was found for *m*-cresol and phenol, with *m*-cresol causing a greater shift, whereas no effect was observed for benzyl alcohol assuming that the blue shifts were caused by hydrophobic association of phenol and *m*-cresol with the peptide. Furthermore, H-NMR experiments were conducted to analyze preservative-induced structural changes of the acyl-peptide around the region of the tryptophan residue. Very interestingly, distinct NMR peaks were observed: In accordance with the fluorescence emission results, up to 2.2% benzyl alcohol did not alter the apparent peak width in the spectrum. The addition *m*-cresol and phenol led to additional peaks around the methyl protons with increasing concentrations (phenol: 0–1.3%; *m*-cresol: 0–0.8%) leading to more observable effects. These spectral changes might be an indicator of structural changes, e.g., peptide-preservative interactions or oligomerization. Moreover, isothermal calibration calorimetry was performed to measure enthalpy changes caused by interactions or changes of the solution behavior. In accordance with the previous findings, phenol and *m*-cresol led to a net endothermic effect of approx. +30 kcal·mol^−1^ indicating that additional heat was involved when the two preservatives were added. No increase in enthalpy was detected for the addition of benzyl alcohol, confirming the assumption that the peptide did not interact with the preservative. Finally, it was shown that the addition of benzyl alcohol increases the physical stability of the formulation induced by salt titration. In detail, higher benzyl alcohol concentrations were found to reduce turbidity values and particle sizes at lower salt concentrations. Because the sensitivity of the benzyl alcohol containing formulation to higher ionic strength decreased, it was concluded that benzyl alcohol does not interfere with the peptide’s structure or stability. [145].

#### 3.2.2. Protein–Preservative Interactions

Maa & Hsu (1996) analyzed the aggregation rate of recombinant human growth hormone (rhGH) at a concentration of 10 mg·mL^−1^ triggered by benzyl alcohol, phenol, and *m*-cresol [33]. Titration microcalorimetry was performed to analyze solute–protein binding, and optical circular dichroism was performed to analyze conformational changes in protein structure in the presence of preservatives. The turbidity measurements showed increased values in the presence of preservatives in the following order: BA (<0.02 OD) < PH (0.04 OD) < CR (0.14 OD). The size of protein aggregates as well as the protein’s stability at higher temperatures followed the same order as the turbidity measurements: The size of the protein species in buffer alone was 5 nm and was increased to 18 nm in solutions with *m*-cresol and 7 nm in solutions with phenol. The protein’s size was observed to be 5 nm in the presence of benzyl alcohol, and thus was not affected by this preservative. Additionally, the degree of aggregation caused by agitation followed the same ranking order as the previously mentioned results of the study. By performing SDS-PAGE for the rhGH samples containing the preservatives, it was suggested the aggregates were non-covalent in nature. Protein–preservative binding analysis via titration microcalorimetry showed that the interactions of all preservatives were of weak hydrophobic nature and the binding enthalpy of the same magnitude. Based on the measured heat involved in the reaction, the reaction enthalpy ΔH° could be calculated. Greater ΔH°-values indicate stronger “binding” interactions. Again, *m*-cresol was reported to have the strongest interactions with rhGH with a reaction enthalpy of ΔH°= −810 cal·mol^−1^ followed by benzyl alcohol (–638 cal·mol^−1^) and phenol (–153 cal·mol^−1^); the fact that phenol has a lower binding enthalpy than benzyl alcohol contradicts previous results of the study; considering the aggregation tendency and thermal stability of rhGH, phenol was always reported to have a more destabilizing effect compared to benzyl alcohol. Finally, the researchers suggested that increasing preservative interactions with the protein would lead to an increased exposure of hydrophobic regions to the surface, causing conformational change.

Zhang et al. (2004) focused on the mechanism of how benzyl alcohol induced aggregation of recombinant human interleukin-1 receptor antagonist (rhIL-1ra) in aqueous solutions [16]. Monomer loss was analyzed using SEC-HPLC. The effect on the tertiary structure was analyzed via UV circular dichroism (CD) and second-derivative UV spectroscopies. Infrared spectroscopy was used to monitor shifts from the protein’s native to partially unfolded state indicated by hydrogen-deuterium exchange rates. Exchange occurs when the hydrogen atoms are exposed to the solvent. Consequently, the exchange rate is accelerated if the equilibrium of protein species shifts from the native to the partially unfolded state. Additionally, thermostability experiments were performed using calorimetry. Last, the thermodynamics of how benzyl alcohol binding to the protein was determined using isothermal titration calorimetry (ITC). Their results demonstrated that the formulation containing 0.9% benzyl alcohol aggregated faster than the negative control containing no AP, e.g., benzyl alcohol caused a monomer reduction of almost 60% compared to approximately 10% in the negative control after two days of incubation at 37 °C. The addition of 0.5 M sucrose to 0.9% benzyl alcohol partially counteracted benzyl alcohol-induced protein aggregation leading to a monomer content of ~50% after 5 days of incubation compared to only 20% in samples including only 0.9% benzyl alcohol. Moreover, CD results showed a decreased signal intensity when benzyl alcohol was added to rhIL-1r indicating a slight perturbation of the protein’s tertiary structure No changes in the secondary structure were observed. Results of second-derivative UV spectroscopy demonstrated that the addition of benzyl alcohol caused an increased exposure of aromatic amino acids to the solvent. Again, this effect could partially be prevented by the presence of 0.5 M sucrose. In accordance with these findings, ANS fluorescence analysis showed a positive correlation between the fluorescence intensity of ANS and the concentration of benzyl alcohol indicating increased levels of partially unfolded protein species at higher preservative concentrations. Hydrogen-deuterium exchange by infrared spectroscopy showed an increased exchange rate in the presence of benzyl alcohol compared to the protein alone. Unfolding experiments with urea or guanidine hydrochloride demonstrated benzyl alcohol did not significantly affect the free energy of unfolding of rhIL-1ra. However, DSC showed that the preservative decreased the apparent melting temperature in a concentration-dependent manner from 56 °C in the negative control to 51, 48, and 44 °C in the presence of 0.9, 1.5, and 2.0% benzyl alcohol. Last, ITC results suggested that the protein–preservative interactions are of hydrophobic nature. With a *K_d_* value in the mM range, it was concluded that the binding affinity was relatively low, and that the preservative’s effect could be attributed to preferential binding. In conclusion, Zhang et al. (2004) hypothesized that benzyl alcohol increased protein aggregation by preferentially binding to partially unfolded protein species via hydrophobic interactions, thereby shifting the equilibrium of the protein towards partially unfolded species. Additionally, they were able to show that the presence of sucrose in the formulation counteracted benzyl alcohol-induced protein aggregation [16].

Roy et al. (2005) studied the effect of benzyl alcohol on the aggregation of human interleukin-1-receptor (rhIL-1ra) antagonist in reconstituted lyophilized formulations [18]. For this, they analyzed the impact of pH, freezing, and formulation variables (pre-lyophilized protein concentration, sucrose as a stabilizing excipient, and NaCl as tonicity agent) on protein unfolding and aggregation during lyophilization and after reconstitution with bacteriostatic water (0.9% w·v^−1^ (83.2 mM) benzyl alcohol). Protein aggregation, by means of SEC, was analyzed during reconstitution and post-reconstitution storage in an aqueous 0.9% w·v^−1^ (83.2 mM) benzyl alcohol solution at room temperature. Some consistent trends could be observed. First, it was shown that the formulation including sucrose showed higher protein stability compared to formulation in buffer alone. The amount of soluble aggregate after reconstitution was higher in bacteriostatic water compared to reconstitution in water with the highest aggregate percentage of 21% observed for the formulation with the lowest protein concentration (15 mg·mL^−1^). Interestingly, PS80 had no effect on protein stability. Last, analysis of protein stability after post-reconstitution storage showed that benzyl alcohol did not affect aggregation when stored at room temperature. In a previous study, Roy et al. (2005) showed that benzyl alcohol induced rapid protein aggregation at elevated temperature (37 °C) due to hydrophobic interactions. Nevertheless, it was suggested that a decrease in incubation temperature from 37 °C to room temperature is already enough to minimize the interactions between rhIL-1ra and benzyl alcohol, resulting in decreased aggregation [18].

Another study by Thirumangalathu et al. (2006) analyzed the effects of pH, temperature, and sucrose on the aggregation tendency of recombinant human granulocyte colony stimulating factor (rhGCSF) triggered by benzyl alcohol [23]. rhGCSF was shown to be stable at very acidic conditions, as low as pH 2, making it a good candidate for studying how pH impacts preservative-induced protein destabilization. Protein stability was analyzed at pH 3.5 and 7.0 and at 25 °C and 37 °C. In all performed experiments, the protein was less vulnerable to benzyl alcohol-induced protein aggregation at pH 3.5 compared to pH 7.0. This phenomenon was attributed to a greater colloidal stability at the lower pH value. Furthermore, SEC-HPLC was used for determination of the monomer content. At 25 °C and pH 7.0, the amount of protein monomers in 0.9% w·v^−1^ (83.2 mM) benzyl alcohol decreased constantly, but the loss was relatively small. At 37 °C, the loss was greatly increased with more than 90% of protein aggregation after the first day of incubation. These results were confirmed by secondary structure analysis which showed a conversion of a native α-helix to a nonnative intermolecular β-sheet induced by the addition of benzyl alcohol, and greater effects were shown at elevated temperatures. Additionally, tertiary protein structure analysis was performed via near ultraviolet (UV) circular dichroism (CD). The results showed that the asymmetric environment of tyrosine and tryptophan was changed in the presence of benzyl alcohol with greater changes at 37 °C compared to 25 °C. IR spectroscopy was performed to measure H-D exchange in real-time to determine the effect of formulation components and conditions on protein structure. In their native structure, the protein’s core is protected from fast H-D exchange. The exchange degree can be accelerated by shifting the equilibrium of protein species to partially unfolded proteins. The presence of benzyl alcohol induced an increased exchange rate at both formulation variables, 25 °C and 37 °C as well as pH 3.5 and 7.0, presuming that the time-averaged conformation was more expanded compared to buffer alone. In all experiments, the destabilizing effect of benzyl alcohol could partially be reversed by the addition of 1.0 M sucrose. Being a preferential-excluded agent from the protein’s surface, the sugar increased the chemical potential of the protein and therefore, decreased the shift to partially unfolded protein species [23].

The study by Hutchings et al. (2013) analyzed the effect of five APs (*m*-cresol, phenol, benzyl alcohol, phenoxyethanol and chlorobutanol) on partial unfolding and aggregation of cytochrome c [22]. Different cytochrome c concentrations were used for different analyses. Analysis methods included antimicrobial efficacy testing, SEC, isothermal incubation experiments to accelerate aggregation kinetics, thermal scanning methods to determine the aggregation temperature, denaturant melts with guanidinium chloride, NMR analysis, and hydrogen exchange experiments. Isothermal incubation experiments at 75 °C and 80 °C showed that all APs increased protein aggregation compared to the negative control in the order CR > PH > BA > PE > CB. In addition, they demonstrated a negative linear correlation between the aggregation temperature and an increasing preservative concentration indicated by a slope (aggregation temperature vs concentration) of −17.5 °C/% v·v^−1^ for *m*-cresol, −14.4 °C/% v·v^−1^ for phenol, −8.8 °C/% v·v^−1^ for benzyl alcohol, −7.0 °C/% v·v^−1^ for phenoxyethanol, and −3.3 °C/% v·v^−1^ for chlorobutanol. Furthermore, they were able to detect an aggregation hotspot of cytochrome c located around the methionine residue at position 80 (Met80). At this position, Met80 was covalently linked to the ferric iron of the heme group. In a previous study, it was shown that the loss of this ligation was the first step in cytochrome c unfolding [156]. By determining the unfolding temperature, Hutchings et al. (2013) could show that APs destabilized this region, and that the destabilization occurred prior to protein aggregation. To further show that APs facilitate unfolding of the aggregation hotspot, they carried out amide HX experiments with 2D NMR examining the amides of Tyr74 and Ile75. The amides of these two residues were reported to exchange with the solvent only if the region around Met80 is completely unfolded [156]. HX exchange experiments by Hutchings et al. (2013) showed that the overall exchange rate was not changed in the presence of the APs, but the local range of aggregation hotspot was significantly accelerated. In detail, the formulation containing only cytochrome c but no AP had an exchange rate constant of 0.57/h for Tyr75 and 0.27/h for Ile75, whereas the rate constant in the presence of phenoxyethanol was 1.73/h for Tyr74 and 1.21/h for Ile75, and in the presence of benzyl alcohol 2.60/h for Tyr74 and 1.19/h for Ile75. The exchange was even more accelerated in the presence of phenol and *m*-cresol as the amide protons exchanged within the dead time of the experiment (~10 min). Last, Hutchings et al. (2013) showed that stabilizing the weakest entity could decrease protein aggregation. In the case of cytochrome c, the oxidation state of iron in the heme was reduced from Fe^3+^ to Fe^2+^ which was predicted to be part of the aggregation hotspot in cytochrome c. The oxidation state increased the aggregation temperature for all five APs in every tested concentration. These results showed that all APs interacted with the same aggregation hotspot, indicating that approaches to stabilize such hotspots might work for all APs [22].

The research group by Hutchings et al. (2013) used the same APs (*m*-cresol, phenol, benzyl alcohol, and phenoxyethanol) analyzed in the previous study [22] but using a different protein interferon, namely α-2a (IFNA2) with a concentration of 10 µM [17]. As mentioned in the previous study, the APs caused an increase in protein aggregation indicated by a more turbid protein solution and a decrease in the monomer content, thus impairing colloidal protein stability. For example, isothermal aggregation kinetics at 50 °C was performed to analyze the monomer content as a function of time. In the absence of preservatives, the monomer content of IFNA2 was 90% after 24 h of incubation, whereas the content decreased to 80% in the presence of phenoxyethanol and benzyl alcohol after 24 h. More drastic effects were seen for *m*-cresol (0.3% v·v^−1^) and phenol detecting no monomer after eight and twelve hours, respectively. Thus, the preservative-induced aggregation rate reported for cytochrome c was in accordance with the preservative-induced aggregation rate of IFNA2 with *m*-cresol accelerating aggregation the most followed by phenol, benzyl alcohol and phenoxyethanol. To further analyze protein aggregation, they measured the absorbance at 450 nm wavelength in dependence of temperature. The 450 nm wavelength was chosen as neither the protein nor formulation components absorbed at this wavelength. Therefore, increased absorbance values could be attributed to the formation of protein aggregates. The results showed that the aggregation temperature was shifted from 63.9 ± 0.9 °C in the absence to 56.7 ± 0.7 °C in the presence of 0.5% phenol indicating that the preservative accelerated protein aggregation. As in their previous study, they could also find a linear correlation between protein aggregation and increasing preservative concentration ranging from 0−0.3, 0−0.5, 0−2.0, and 0−2.0% v·v^−1^ for *m*-cresol, phenol, benzyl alcohol, and phenoxyethanol, respectively. The slopes of the linear fit, representing the efficiency of an AP to cause protein aggregation, were –22.1, –14.3, –10.5, and –8.2% v·v^−1^ for *m*-cresol, phenol, benzyl alcohol, and phenoxyethanol, respectively. Thus, the aggregation rate of phenoxyethanol was approximately 3 times smaller compared to *m*-cresol. As all their experiments with IFNA2 were in accordance with their previous investigations with cytochrome c, the authors suggested that the general mechanism of preservative-induced protein aggregation may be independent of the protein species depending only on the nature of the preservative [17].

Torosantucci et al. (2018) used NMR spectroscopy for studying the interactions between a model protein and phenol [21]. For analyzing the binding behavior, they applied the same method as described by Heljo et al. (2015) [19]. Different protein concentrations (0, 5, and 30 mg·mL^−1^) and preservative concentrations (0, 0.3, and 3 mg·mL^−1^ (3.2 mM and 31.9 mM, relatively)) were tested. A significant decrease in the diffusion coefficient from ~1.1 × 10^−9^ to 0.5 × 10^−9^ m^2^·s^−1^ was observed in formulations containing the protein compared to placebo formulations. It was shown that formulations with the highest protein concentration (30 mg·mL^−1^) showed the greatest percentage of bound phenol, being 53%. In comparison, 38% of phenol was bound in formulation containing 5 mg·mL^−1^ protein. Importantly, the decreased diffusion coefficient was not a result of a changed viscosity because the viscosity value of the sample containing 5 mg·mL^−1^ protein did not considerably differ from the values of the placebo. It was also shown that increasing the phenol concentration led to a slightly reduced binding percentage possibly due to saturation of the binding sites leading to the assumption that the percentage of bound phenol increases at higher protein concentration. Last, the antimicrobial efficacy of the active formulation and a placebo was analyzed. In accordance with the API–phenol interactions found by the NMR analysis, the active formulation failed to meet any criteria of the antimicrobial efficacy test of the Ph.Eur., whereas the placebo formulation has reached criterion B. [21]. These results confirm that a preservative is only active in its free form, and binding to other components reduces its efficiency.

#### 3.2.3. Antibody—Preservative Interactions

Gupta & Kaisheva (2003) studied the effect of benzyl alcohol, chlorobutanol, methylparaben, propylparaben, phenol, and *m*-cresol on the stability of a humanized monoclonal antibody in a liquid formulation [5]. The formulation contained 10 mg·mL^−1^ antibody in histidine buffer at pH 6.0, including PS80 and NaCl. The tested preservative concentrations were: Benzyl alcohol (0.75%, 0.5%, and 0.1%), chlorobutanol (0.2%, 0.1%, and 0.05%), methylparaben (0.1%, 0.05%, and 0.01%), propylparaben (0.01%, 0.0075%), phenol (0.5%, 0.1%), and *m*-cresol (0.3% and 0.1%). For analyzing the preservative’s effect on protein stability, the liquid formulations were incubated at 50 °C for two days. The greatest destabilizing effect was seen for *m*-cresol and phenol. The presence of pharmaceutically relevant *m*-cresol concentrations (0.1–0.3%) led to protein precipitation, and the formulation with 0.5% phenol decreased the monomer content to ca. 20% after two days of incubation at 50 °C. Therefore, phenol and *m*-cresol were excluded from further analysis. The protein was the most stable in formulations containing methylparaben and propylparaben. After 2 d/50 °C, the monomer content was still high at 91.3% and 85.3%, respectively. Analysis of the denaturation temperature showed a decrease by 1.5 °C induced by benzyl alcohol and chlorobutanol compared to the control sample whereas methyl- and propylparaben had no impact on the temperature. The antimicrobial efficiency of benzyl alcohol, chlorobutanol, methylparaben, and propylparaben used as single preservatives and in combination was tested. Formulations including 0.75% benzyl alcohol met the criteria of the USP and Ph.Eur. (criterion B) antimicrobial efficiency tests, and formulations including 0.2% chlorobutanol, 0.1% methylparaben and 0.01% propylparaben showed antifungal activity. However, increased preservative concentrations, especially in the case of benzyl alcohol, led to a significant decrease in the protein’s monomer content. Therefore, it was analyzed in silico whether sufficient antimicrobial protection can be obtained by combining preservatives at lower concentrations. Indeed, synergistic effects were observed, with the greatest effect against bacteria found for the combination of methylparaben with propylparaben, and the strongest negative effect for benzyl alcohol with methylparaben. Synergistic activity against bacteria was also obtained by combining benzyl alcohol with propylparaben, and the combination of chlorobutanol with methylparaben showed increased antimicrobial activity against fungi and bacteria. The combination of benzyl alcohol with chlorobutanol increased their efficiency against fungi but decreased their efficiency against bacteria. The in-silico experiments showed that combining preservatives could not only increase the antimicrobial protection (for more details, see Section 2.1.5), but could also reduce the preservative-induced protein destabilization as lower concentrations would be required [5].

Kaja et al. (2011) studied the compatibility of Avastin, a recombinant humanized anti-VEGF immunoglobulin G1 antibody intended for nasal administration, and benzalkonium chloride in standard high-density polyethylene (HDPE) bottles [157]. For their study, Avastin was diluted to 10 mg·mL^−1^ with 0.9% sterile normal saline containing benzalkonium chloride at a concentration of 0.013%. Samples were aliquoted into polyethylene vials and stored at either −60 °C or 4 °C for two weeks. Native polyacrylamide gel electrophoresis (PAGE) was performed in TRIS-glycine buffer to determine antibody degradation or aggregation. Neither smaller bands nor smearing indicating the presence of degraded antibody species nor larger bands indicating aggregated antibody species were detected after two weeks of storage. As native PAGE might reach detection limit for smaller degradation products, sodium dodecyl sulfate (SDS)-PAGE was also performed, but SDS-PAGE analysis could also not reveal any suspicious bands. The results of both analyses indicated that the protein neither aggregated nor was degraded after prolonged storage in a benzalkonium chloride-containing diluent. Moreover, enzyme-linked immune-sorbent assay (ELISA) was carried out to determine the concentration of Avastin which was calculated to be 25 ng. Avastin samples were analyzed at the beginning (T_0_) and after storage (T_14_). The mean concentrations were 23.0 ± 1.7 ng for T_0_ and 22.4 ± 2.4 ng for T_14_, showing no statistically significant difference (n = 4, *p* = 0.835). Finally, isoelectric focusing was performed to analyze subtle changes on the electrochemical properties of the antibody. The isoelectric point was found to be approximately at pH 8.3. No changes in density were detected at T_0_ and T_14_, and no additional bands were seen after subsequent separation by SDS-PAGE. Thus, it was shown that the electrochemical properties of Avastin remained unchanged after dilution and storage. Overall, the performed experiments revealed no statistically significant changes between T_0_ and T_14_, indicating that the dilution in benzalkonium chloride-containing sterile normal saline and prolonged storage had no impact on the stability of Avastin [157].

Arora et al. (2017) investigated the impact of *m*-cresol, phenol, benzyl alcohol and phenoxyethanol on the conformational stability, aggregation tendency, and backbone flexibility of an IgG1 monoclonal antibody, mAb-4 [15]. mAb-4 was added to the formulation (20 mM citrate-phosphate buffer at pH 6.0, with 100 mM NaCl) at a concentration of 1 mg·mL^−1^. The amount of added preservatives was the same for all formulations (53 mM). Extrinsic fluorescence spectroscopy and DSC results demonstrated that all preservatives decreased the conformational stability of a monoclonal antibody compared to a negative control in correlation to their log*P* values. For example, *m*-cresol (log*P*: 1.96) showed the greatest decrease in the melting temperature $∆T_{\mathrm{Onset}}$ (−7.7 °C) followed by phenol ($∆T_{\mathrm{Onset}}$: −4.6 °C; log*P*: 1.46), phenoxyethanol ($∆T_{\mathrm{Onset}}$: −3.3 °C; log*P*: 1.2) and benzyl alcohol ($∆T_{\mathrm{Onset}}$: −2.8 °C; log*P*: 1.05). Moreover, they carried out an accelerated storage stability study at 50 °C for 28 days using SEC analysis for comparing the content of monomers, aggregates, and fragments of the antibody. A significant increase of aggregates and fragments was found in formulations containing preservatives after 28 days of storage. The ranking order of formed aggregates was the following: CR (15%) > PH (9%) > PE (6%) > BA (4%). As before, the order correlated with the preservatives’ log*P* values. The antibody’s flexibility was studied via mass spectrometry by evaluating the hydrogen-deuterium exchange rate. Deuterium uptake was measured every 30 sec and converted to percent flexibility. Hydrogen exchange profiles of formulations containing the preservatives were compared to the negative control to analyze the impact of the preservatives on the local backbone flexibility of the monoclonal antibody. Overall, an increase in local flexibility was induced by all four preservatives, but not all increases were statistically significant. However, a significant increase in local backbone flexibility was reported for one specific region indicating that the existence of an aggregation hotspot. The results correlated with the preservatives’ effect on protein stability and aggregation tendency, with more hydrophobic APs causing greater exchange rates [15].

Table 14 summarizes the results of the beforementioned studies regarding the preservative-induced protein aggregation rate, including peptides/proteins in general, and antibodies in particular.

### 3.3. Comparison of Liquid and Lyophilized Multi-Dose Biological Formulations Regarding the Biological Compound’s Stability and Microbial Protection

Multi-dose biological formulations exist in several forms, e.g., as injections, ointments, or lyophilized powders. The most commonly used are injection solutions and lyophilized powders for solution. The main difference between these forms is that in liquid formulations, the preservative is added during the manufacturing process being part of the formulation, whereas in lyophilized formulations, it is only added to the formulation during reconstitution by the reconstitution medium. Thus, the protein gets in contact with the preservative only after being reconstituted with the diluent. This reduces potential long-term incompatibilities between the preservative and the biological compound as just in use stability handling times are required. The drawbacks are that lyophilizates need to be reconstituted prior to application, self-administration is unlikely, and the manufacturing costs are higher. Additionally, the AP cannot be added prior to lyophilization due to volatility during the drying cycle [18]. To our knowledge, there are no licensed formulations on the market in which the preservative is added prior to lyophilization. On the contrary, the probability that the preservative interacts with the protein and thereby decreases its stability is increased in liquid formulations.

### 3.4. How to Avoid Preservative-Induced Protein Destabilization and Aggregation?

Which aspects should be considered for a successful multi-dose biological formulation? It must demonstrate sufficient microbial protection without destabilizing the active pharmaceutical ingredient. Therefore, it is of great importance to stabilize the biological compound against preservative-induced aggregation and destabilization. In the following, effective stabilizers and/or formulation parameters are discussed.

As described earlier (Section 3.1), preferentially excluded excipients, e.g., sucrose and trehalose, successfully inhibited AP-induced protein aggregation as shown from previous studies [16,18,23]. According to the presented reports, such excipients minimize the amount of partially unfolded protein species as they are excluded from the protein’s surface, and therefore stabilize the compact, native state. This stabilizing action leads to an increase in the chemical potential of the protein. As the extent of preferential exclusion and chemical potential is directly linked to the surface area of the protein to the solvent, the degree of exposure is greater for partially and fully unfolded proteins compared to native proteins. Thus, preferential excluded excipients seem to stabilize the protein by shifting the equilibrium towards fully folded proteins [23]. In previously described studies, sucrose could decrease BA-induced protein aggregation of a lyophilized formulation after reconstitution in bacteriostatic water compared to formulations without sucrose [18]. In addition, Rodriguez-Silva et al. (1999) investigated the “positive” stabilizing effect of 1.0 M sorbitol against phenol-induced antibody aggregation [154]. Furthermore, Yoshizawa et al. (2018) reported that the co-solvent trimethylamine N-oxide (TMAO) showed promising effects as a suppressor of BA-induced IgG1 aggregation showing higher stabilizing effects compared to other additives like trehalose, xylitol, or arginine [155].

Additionally, finding an appropriate pH is crucial to counteract AP-induced protein aggregation. It is favorable to find a pH value at which intermolecular charge-charge repulsion are increased which, in return, minimizes intermolecular interactions and aggregation of the biological compound [16,23]. Obviously, this strategy is only beneficial if the pH itself does not impact the compound’s degradation as it would counteract the advantages of intermolecular repulsion. For instance, Thirumangalathu et al. (2006) found that BA did not increase rhGCSF aggregation at pH 3.5 but led to increased aggregation and precipitation at pH 7.0 [26]. They attributed that outcome to the positive charge of the protein at lower pH leading to beneficial intermolecular repulsions between protein molecules. Maa & Hsu (1996) observed the same correlation as Thirumangalathu et al. (2006) [33].

Temperature is another factor which impacts AP-protein interactions. Elevated temperatures were shown to accelerate protein aggregation in the presence of preservatives as the amount of partial unfolded proteins is increased at higher temperatures, and preservatives are attributed to preferentially bind to and stabilize such protein species [16,18,23].

Moreover, Evers et al. (2019) showed promising results with their in-silico models predicting physicochemical properties of peptides and their binding behavior to excipients. They could rationalize pH-dependent peptide aggregation induced by the addition of phenol and *m*-cresol. Additionally, it was possible to find aggregation hotspots by searching for hydrophobic portions on the peptide surface that might interact with the preservative. By identifying these regions, they could exchange hydrophobic residues, e.g., tryptophan, for hydrophilic ones, e.g., lysine, leading to an increased charge and preventing conformational collapsing induced by the addition of preservatives. Thus, Evers et al. (2019) recommend using in silico models, e.g., sequence analysis, molecular models, and computed physicochemical descriptors, to imitate the effect of pH or formulation components on the physical and chemical stability of the protein [20].

To sum up, AP-induced destabilization and aggregation of the biological compound can be counteracted by (i) the addition of preferentially-excluded stabilizer (e.g., sucrose, trehalose, TMAO), (ii) increasing intermolecular charge-charge repulsion between proteins, (iii) identifying and stabilizing aggregation hotspots, (iv) avoiding the exceedance of the temperature, (v) choosing appropriate process materials that reduce or avoid adsorption to or diffusion into the contact material interface (e.g., fluoropolymer tubing), (vi) choosing primary packaging material components avoiding preservative adsorption and loss, such as tear-off seals consisting of three aluminum layers, or (vii) choosing a lyophilized instead of a liquid formulation.

## 4. Outlook: Alternatives to Commonly Used Preservatives

Contamination of pharmaceutical products by microorganisms is a critical issue and thus, has led to the development of chemical preservatives. However, safety concerns for some preservatives, in particular benzalkonium chloride, benzyl alcohol, and parabens, were reported in the past [66,67,68,71,78,79,80,81,82,83,84,85]. Despite the reported health risks, it is also possible that microorganisms will develop a resistance against commonly used preservatives. For example, Close & Nielsen (1976) studied the effect of adding the bacterial strain *Burkholderia cepacian* to an oil-in-water emulsion preserved by methylparaben and propylparaben. Very interestingly, these bacteria were not only able to hydrolyze methyl- and propylparaben but could also use propylparaben as their carbon and energy source. Regarding the degradation pathway, it was postulated that parabens were degraded by a bacterial esterase [37]. Thus, researchers have been looking for efficient alternatives being less harmful compared to conventional APs with similar or improved antimicrobial activity [158,159,160].

Over the last years, alkyl glycosides (AG) have captured attention in the scientific community as alternatives. They are composed of naturally occurring, renewable sources, namely a carbohydrate linked to an alkyl chain (Figure 3). Alkyl glycosides have been evaluated as surfactants in formulation before [161,162,163], and may have some antimicrobial activity, too [158,160] coming with several advantages. For instance, their synthesis is simple and requires only a few reaction steps [158]. Moreover, they are considered “generally recognized as safe” (GRAS) due to their biodegradability and -compatibility [158,163].

Alkyl glycosides act on the cell membrane of microorganisms. Being incorporated into the bilayer, they either cause membrane modification or disruption, leading to cell lysis [158,160]. Additionally, enzymatic inhibition assays revealed that AGs inhibited key maltose metabolism enzymes like glucoamylase or α-amylase [160]. The efficacy of AGs depends on the balance between the hydrophilic sugar core and hydrophobic alkyl chain. Two studies analyzed the antimicrobial efficacy of alkyl glycosides. Bilkovà et al. (2015) showed that alkyl glycosides having dodecyl carbon chain length were the most efficient with MIC values in the micromolar range. Against *S. aureus*, *O*-mannosides led to the greatest efficacy, whereas the yeast *C. albicans* was the most susceptible to thiomannosides [158]. In a different study, Marçon et al. (2013) compared the effect of several alkyl-*β*-*O*-oligomaltosides with a monosaccharide control (MeG1), propylparaben, and an antibiotic (ciprofloxacine). The tested microorganisms were Gram-positive and -negative bacteria as well as fungal strains. Besides MeG1, all tested components demonstrated antimicrobial activity. Therefore, it was suggested that monosaccharides had only minimal to negligible antimicrobial activity. In many cases, the efficacy of alkyl glycosides was even higher than the one exhibited by propylparaben. For maltosides, methyl derivatives showed a greater efficacy against bacteria than dodecyl derivatives. Analyzing the efficiency against fungal strains, dodecyl derivatives demonstrated greater activity. Comparing methyl compounds, the maltosyl derivative showed smaller MIC values, whereas for dodecyl compounds, the opposite was the case [160]. The combination of DoG3/MeG3 exhibited a synergistic antimicrobial effect against *E. coli* and *S. aureus.* Taken together, alkyl glycosides show promising antimicrobial efficacy. They are hydrolyzed to sugars and long-chain alcohols within the body, and thus exhibit low toxicity [158]. However, their mode of action and pharmaceutical usage remain an open field requiring further investigations regarding their potential as APs.

## 5. Licensed Multi-Dose Protein and Peptide Therapeutics

Besides the scientific background, this review also gives an overview about multi-dose peptide, protein, and antibody formulations on the market. An overview about the distribution of licensed products depending on the used AP is depicted in Figure 4, Figure 5 and Figure 6. In Figure 4, market products are considered that contain one preservative. Figure 5 contains preservative combinations of two or three preservatives. Mostly, this includes the combination of methylparabens and propylparabens mainly due to the low solubilities of the two. Figure 6 shows market products that contain bacteriostatic water. These are lyophilized products that are reconstituted with bacteriostatic water. Further, 12 products contain benzyl alcohol, followed by *m*-cresol with six matches.

Regarding market products containing one preservative, as seen from Figure 4, *m*-cresol and phenol are the mostly used APs with 62 and 27 found, licensed biological formulations, relatively. This is mainly because they are the preservatives of choice for formulations, including insulin derivatives. As mentioned above, the addition of phenol and chlorobutanol can lead to a more stable, compact structure of the insulin molecule [57] leading to their increased number in multi-dose insulin formulations. Benzyl alcohol is also found in many formulations (26 licensed drugs) containing growth hormone, human chorionic gonadotropin, and interferon alpha2-1a as an API. Benzalkonium chloride is included in 17 formulations, followed by chlorobutanol (22 formulations), methylparaben (3 formulations). Phenoxyethanol is found in three biological formulations. However, it was demonstrated that its addition caused less protein aggregation compared to more frequently used APs in previous studies [15,17,22]. Thus, it might be a promising candidate for the development of new multi-dose formulations, too.

The preservative combinations as shown in Figure 5 are essentially *m*-cresol/phenol combinations (87% of matches) followed by methlyparaben/propylparaben combinations (9 matches found). *m*-Cresol/phenol combinations are mainly used for subcutaneous administration in insulin containing formulations, whereas methylparaben/propylparaben combinations are for the nasal, topical, or ophthalmic route. Finally, in Figure 6, market products with bacteriostatic water used for reconstitution of lyophilized products are shown.

Figure 7 shows the distribution of the APIs in marketed multi-dose biological formulations. As can be seen, insulin formulations are by far the most frequently marketed multi-dose drugs.

Very importantly, benzyl alcohol is the only AP which is used in a multi-dose antibody formulation, including the API trastuzumab. The dosage form is a lyophilized powder for injection. Benzyl alcohol is not part of the lyophilized powder, but of the reconstitution medium. Several companies manufacture multi-dose formulations of trastuzumab. For example, the description of Herceptin, manufactured by Genentech, Inc [164] is shown in Table 15.

A case study showed that benzyl alcohol was able to stabilize a pharmaceutical formulation comprising a bispecific antibody in both solution and lyophilized state [165]. In the described formulation, the stabilizing effect was attributed to the preservative’s effect on the second binding domain. This is of great interest as APs were almost always attributed to destabilize proteins. Even though benzyl alcohol leads to partial unfolding of certain regions, it could prevent the formation of dimers or multimers in the frozen state. In the presence of benzyl alcohol, the percentage of high molecular weight species was less than 1% out of the total amount of bispecific antibody in a frozen solution. Indeed, it does stabilize the antibody as its binding regions overlapped with regions that were probably involved in dimer formation, e.g., CD19xCD3, BCMAxCD3, PSAMxCD3, CD33xCD3, or EGFRvIIIxCD3. Thus, benzyl alcohol-induced regional unfolding prevented antibody aggregation by conversion to a monomer antibody species. As concluded by Abel et al. (2018)*,* these findings come with several advantages. First, continuous infusion over longer intervals would be possible as the preservative ensured microbial protection. Second, during storage in a frozen state, benzyl alcohol may act as a stabilizing agent, and after dilution, can function as a preservative, too [165].

So far, only one multi-dose antibody formulation containing the antibody trastuzumab exists on specific markets [166]. According to the Herceptin initial marketing authorization documents published by the European Medicines Agency (EMA) [167], “a non-acceptance of the submitted multidose finished product formulation which originally contained benzyl alcohol after reconstitution” is not in compliance with the Ph. Eur. It is re-emphasized that “the single dose vial of 150 mg was used for clinical trials outside the US. However, in the dossier originally submitted Herceptin was presented as a multidose formulation of 440 mg trastuzumab to be reconstituted with 20 mL of Bacteriostatic Water for Injection, containing 1.1% benzyl alcohol to yield a multi dose formulation at 21 mg/mL trastuzumab. As the use of preservative was contrary to the Ph. Eur. requirements, the applicant, following a CPMP (Committee for Proprietary Medicinal Products) request, changed to 150 mg single dose vials to be reconstituted with sterile water for injections without preservative.” The Summary of Product Characteristics published by the EMA (last updated 28.07.2021) describes Herceptin as a 150 mg powder for concentrate for solution for infusion, formulated with histidine/trehalose/polysorbate 20 which is reconstituted with sterile water for injection to a final protein concentration of 21 mg·mL^−1^.

However, the product monograph of ^Pr^HERCEPTIN^®^ trastuzumab for injection describes 440 mg trastuzumab/vial, sterile powder for intravenous infusion only [168], a reconstitution medium composed of “Bacteriostatic Water for Injection (BWFI) supplied with HERCEPTIN (trastuzumab) contains 1.1% benzyl alcohol” (Table 15). Due to the presence of benzyl alcohol, it pointed to the WARNINGS AND PRECAUTIONS section. For the excipient benzyl alcohol, it is noted that “Benzyl alcohol, used as a preservative in BWFI, has been associated with toxicity in neonates and children up to 3 years old. For patients with a known hypersensitivity to benzyl alcohol (the preservative in BWFI), reconstitute HERCEPTIN with Sterile Water for Injection (SWFI). Use SWFI-reconstituted HERCEPTIN immediately and discard the vial.” It is emphasized that “each vial of HERCEPTIN should be reconstituted with 20 mL of BWFI, containing 1.1% benzyl alcohol, as supplied, to yield a multi-dose solution containing 21 mg/mL trastuzumab […]. If the patient has a known hypersensitivity to benzyl alcohol, HERCEPTIN **must** be reconstituted with Sterile Water for Injection”.

However, other researchers tested several antibody formulations regarding their compatibility with APs [5,165,169,170,171,172].

## 6. Conclusions

This review gives a detailed overview of currently used preservatives for biologics, such as protein, peptide, and especially antibody drugs. It covers main aspects that should be considered when developing appropriate formulations and choosing an antimicrobial preservative, including its microbial spectrum, physico-chemical properties, and interactions with the active ingredients, packaging and process material interfaces, consumables, and excipients. Additionally, it provides an overview of the antimicrobial effectiveness tests a formulation must fulfill to meet the criteria of the corresponding health authorities. We have summarized all relevant published case studies regarding the interaction of a biological compound with a preservative, providing insights into the current knowledge in the field, including performed methods, stabilizing and destabilizing effects of preservatives, and how to counteract the instabilities of the biological compound caused by the preservative. Based on these publicly available data on this topic, we have provided some potential recommendations and guidance to formulation scientists to direct their development efforts. Last, the review delivers an up-to-date summary of currently licensed multi-dose biological formulations with regard to the used preservatives, administration route, and formulation excipients.

## Figures and Tables

**Figure 1 pharmaceutics-15-00563-f001:**
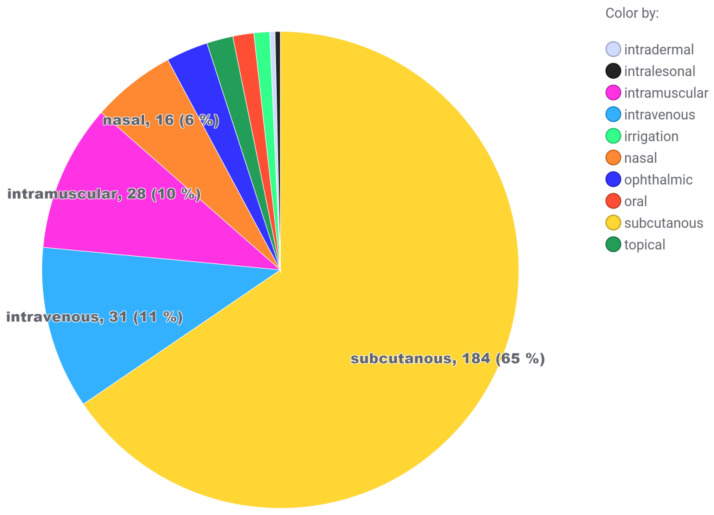
Number and percentage distribution of preservative-containing market products (protein and peptides) in relation to the application route. This figure includes products containing one preservative as well a preservative combinations and products with antibacterial water.

**Figure 2 pharmaceutics-15-00563-f002:**
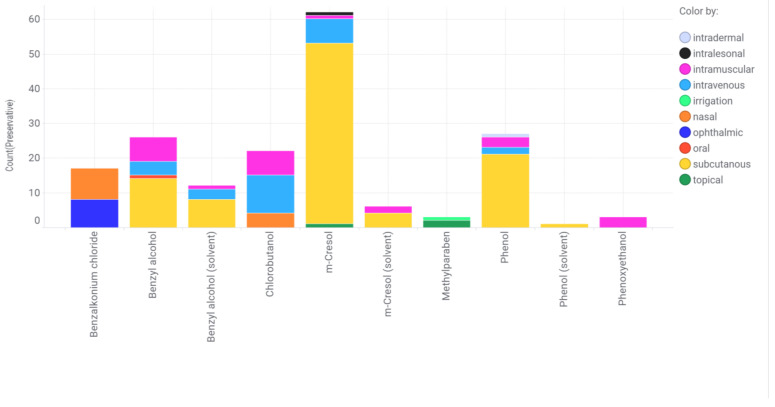
Preservative distribution in dependence on the route of administration. The values are given as the number of correlations found in market products (protein and peptides). Combinations of preservatives and hits from antibacterial water were not taken into account.

**Figure 3 pharmaceutics-15-00563-f003:**
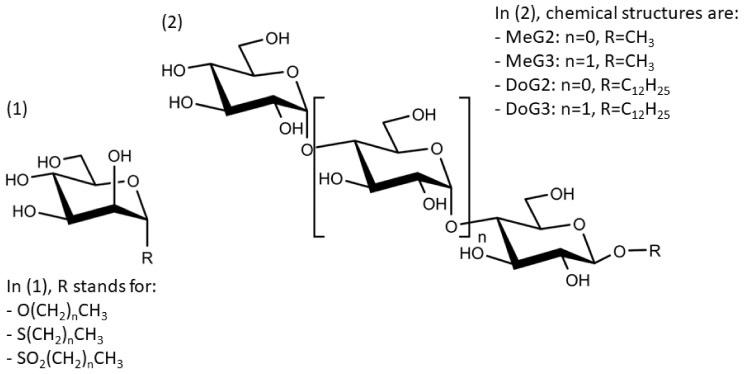
(1) D-mannosides as tested by Bilkovà et al. (2015) with R standing for either an O-glycosidic (O(CH_2_)_n_CH_3_), S-glycosidic (S(CH_2_)_n_CH_3_) or sulfonyl (SO_2_(CH_2_)_n_CH_3_) linkage. The alkyl chain length ranges from 6 to 20 carbon atoms [158]. (2) Alkyl-β-O-oligomaltosides as tested by Marçon et al. (2013) with R standing for either a methyl or dodecyl residue [160].

**Figure 4 pharmaceutics-15-00563-f004:**
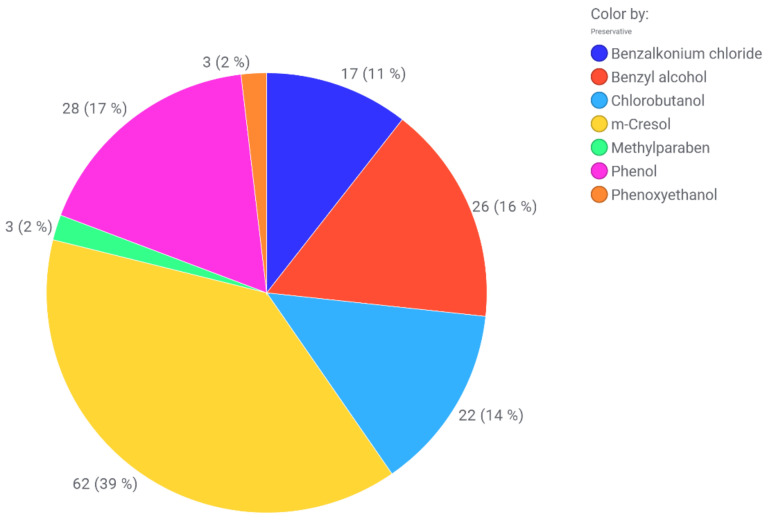
Preservative distribution in licensed protein, peptide, or antibody formulations. Of the currently marketed multi-dose protein formulations, *m*-cresol is used the most (39%) followed by phenol (17%). This is mainly because they are the preservative of choice in insulin formulations where they were reported to stabilize the insulin hexamer. The usage of the remaining preservatives follows the order benzyl alcohol (16%), benzalkonium chloride (11%), chlorobutanol (14%), methylparaben (2%), and phenoxyethanol (2%). Note: The use of rounded percent values results in a total of 101%.

**Figure 5 pharmaceutics-15-00563-f005:**
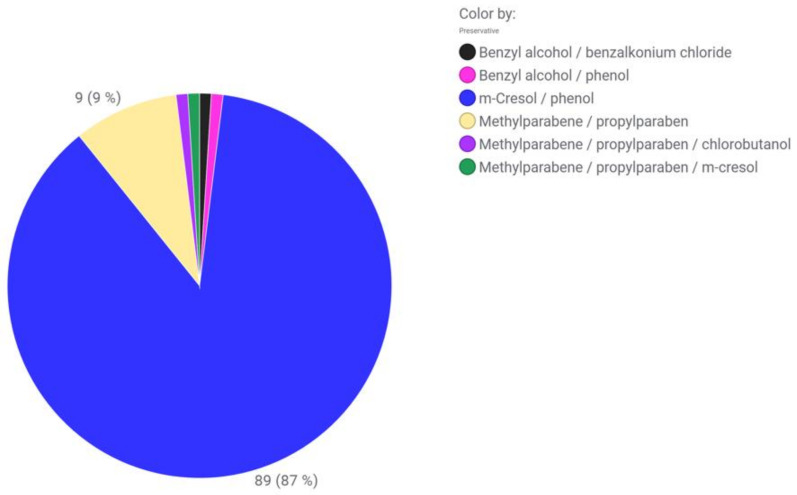
Distribution of preservative combinations containing 2 or 3 preservatives in licensed protein, peptide, or antibody formulations.

**Figure 6 pharmaceutics-15-00563-f006:**
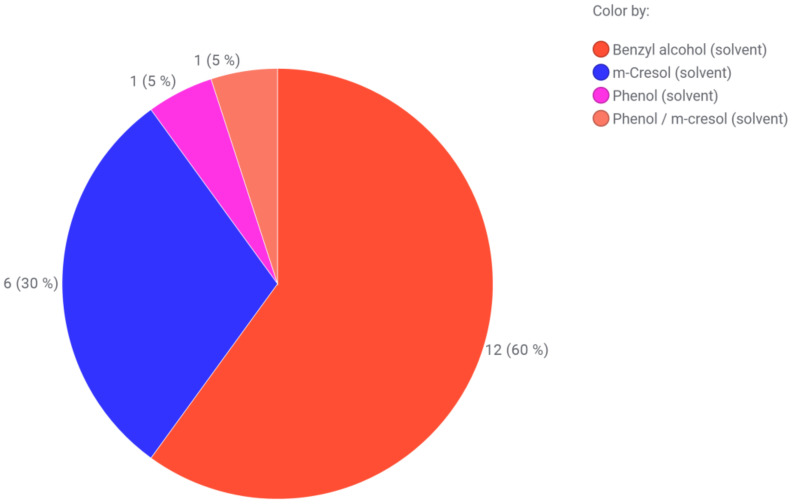
Preservative distribution in licensed protein, peptide, or antibody formulations containing bacteriostatic water.

**Figure 7 pharmaceutics-15-00563-f007:**
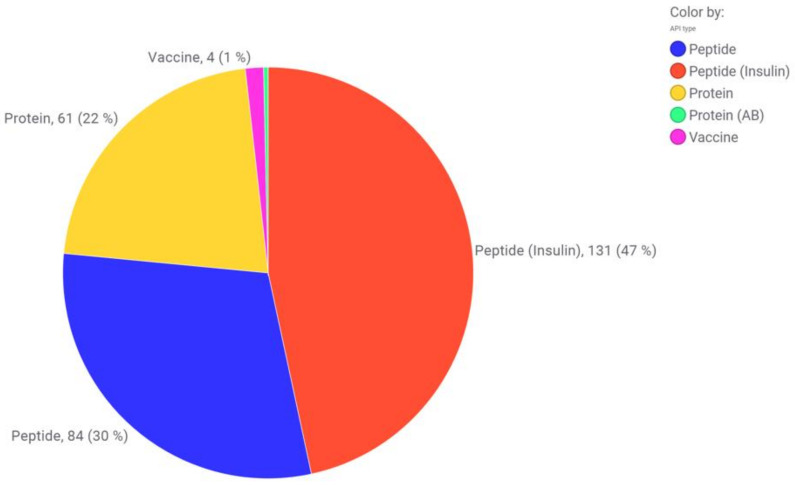
Distribution of market products related to APIs containing preservatives. A peptide length of 100 amino acids was used to discriminate between protein and peptide. 47% of the market products found contain insulin. Of minor importance are vaccines (4 matches) and antibody (AB) formulations (1 match).

**Table 1 pharmaceutics-15-00563-t001:** Structural formulae, molecular weight [g·mol^−1^], and log*P* values of commonly used antimicrobial preservatives. Abbreviations: R = mixtures of alkyls.

Antimicrobial Preservative	Chemical Structure	Molecular Weight (MW)/g∙mol^−1^	log*P* (Hydrophobicity)
Benzalkonium chloride	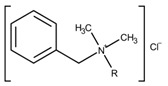	Average weight: 360.00 ^a^	R = C_12_H_25:_ 9.98 ^a^R = C_14_H_29_: 32.9 ^a^R = C_16_H_33_: 82.5 ^a^
Benzyl alcohol	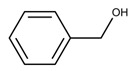	108.14 ^a^	1.05 ^b^
Chlorobutanol	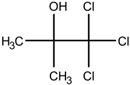	177.46 ^a^	2.03 ^c^
*m*-Cresol	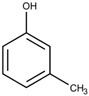	108.14 ^a^	1.98 ^b^
Methylparaben	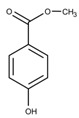	152.15 ^a^	1.96 ^d^
Phenol	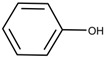	94.11 ^a^	1.48 ^b^
Phenoxyethanol	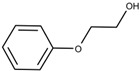	138.16 ^a^	1.2 ^e^
Propylparaben	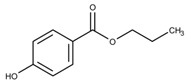	180.20 ^a^	3.04 ^f^

^a^ [14]; ^b^ [32]; ^c^ (PubChem CID 5977, 2021); ^d^ (PubChem CID 7456, 2021); ^e^ (PubChem CID 31236; 2021); ^f^ (PubChem CID 7175, 2021).

**Table 2 pharmaceutics-15-00563-t002:** Classification of preservatives based on their mechanisms and activity sites in microbial cells. Information about the chemical structure important for the antimicrobial activity is added if known. Abbreviation: NI= no information.

Group	Target	Preservative	Chemical Structure Affecting Antimicrobial Activity
1	Cytoplasmic membrane	Alcohols	Bulky and hydrophobic structures, e.g., benzene rings, might increase the interference with the membrane
		Parabens	Increasing chain length positively affects antimicrobial activity
		Phenol	NI
		Quaternary ammonium compounds	Positive charge important for interaction with membrane; alkyl residues important for membrane penetration
2	Separation of daughter cells/cell wall	Phenol	NI
3	Amino acid metabolism	Phenol	NI
4	Protonmotive force	Phenoxyethanol	NI
		Parabens	Carbon chain length

**Table 3 pharmaceutics-15-00563-t003:** Antimicrobial Efficacy Test of the European Pharmacopoeia. The log10 reduction represents the antimicrobial efficacy that must be achieved to meet criteria A and B. The values differ depending on the administration route [10].

			log_10_ Reduction				
Administration Route	Micro- Organism	Criterion	Time criteria				
			**6 h**	**24 h**	**7 d**	**14 d**	**28 d**
Category 1 Parenteral, eye, intrauterine, intramammary	Bacteria	A	2	3	-	-	NR
		B	-	1	3	-	NI
	Fungi	A	-	-	2	-	NI
		B	-	-	-	1	NI
Category 2 Ear, nasal, cutaneous, inhalation			**2 d**	**7 d**	**14 d**	**28 d**	
	Bacteria	A	2	3	-	NI	
		B	-	-	3	NI	
	Fungi	A	-	-	2	NI	
		B	-	-	1	NI	
Category 3 Oral, oromucosal, rectal			**14 d**	**28 d**			
	Bacteria	-	3	NI			
	Fungi	-	1	NI			

NI: no increase in number of viable microorganisms compared to the previous reading. NR: no recovery. More detailed information about the methods is found in the corresponding Ph. Eur.

**Table 4 pharmaceutics-15-00563-t004:** USP43-NF38 criteria for antimicrobial efficacy tests depending on the administration routes. The log_10_ reduction represents the antimicrobial efficacy that must be achieved to reach efficacy standards. The values differ depending on the administration route [8].

		log_10_ Reduction		
Administration Route	Microorganism	Time Criteria		
		**7 d**	**14 d**	**28 d**
Category 1	Bacteria	1.0	3.0	NI from 14 d count
	Yeast and molds	NI from initial count at 7 d, 14 d, and 28 d		
Category 2		**14 d**	**28 d**	
	Bacteria	2.0	NI from 14 d count	
	Yeast and molds	NI from initial count at 14 d and 28 d		
Category 3		**14 d**	**28 d**	
	Bacteria	1.0	NI from 14 d count	
	Fungi	NI from initial count at 14 d and 28 d		
Category 4	Bacteria, Fungi, and Molds	NI from initial count at 14 d and 28 d		

NI: No increase defined as not more than (NMT) 0.5 log_10_ units more than the value to which it is compared to. More information is found in the corresponding USP.

**Table 6 pharmaceutics-15-00563-t006:** pH spectrum for optimal antimicrobial activity of preservatives (left) [12,14] and pH range [13,14,74] found in already licensed protein formulations (right).

Preservative	pH Range	pH Range in Licensed Protein Formulations
Benzalkonium chloride	4–10	3.0–6.2
Benzyl alcohol	<8	6.0–8.0
Chlorobutanol	<5 almost inactive at pH > 5.5	3.0–5.0
*m*-Cresol	<9	3.8–7.8
Phenoxyethanol	3–10	NI
Parabens	4–8	NI
Phenol	< 9	4.2–7.8

NI: no information.

**Table 7 pharmaceutics-15-00563-t007:** Solubility (20 °C) of antimicrobial preservatives in water and ethanol (95%) [14,24]. Units for solubility are reported according to Rowe et al., 2006 [14].

Preservative	Solubility in Water	Solubility in Ethanol (95%)
Benzalkonium chloride	1 in 1.5	1 in 2.5
Benzyl alcohol	1 in 25	Miscible in all proportions
Chlorobutanol	1 in 125	1 in 1
*m*-Cresol	1 in 50	1 in (1–10)
Methylparaben	1 in 400	1 in 3
Phenoxyethanol	1 in 43	miscible
Phenol	1 in 15	1 in less than 1
Propylparaben	1 in 2500	1 in 1.1

**Table 8 pharmaceutics-15-00563-t008:** List of LD_50_ according to the European Chemical Agency (ECHA) and European Medicines Agency (EMA).

Preservative	Route	LD_50_ / mg·kg^−1^ Bodyweight
Benzalkonium chloride	-	-
Benzyl alcohol	Inhalation (rats)	>4178
	Oral (rats)	1620
Chlorobutanol	Oral (rats)	510
*m*-Cresol	Oral (rats)	242
	Dermal (rabbits)	2050
Methylparaben	Oral (rats)	2100
Phenol	Oral (rats)	340–540
	Oral (humans)	140–290
	Dermal (female rats)	660
Phenoxyethanol	Oral (female rats)	1840
	Oral (male rats)	4070
	Dermal (rabbits)	2214
Propylparaben	Oral (rats)	>5000

**Table 9 pharmaceutics-15-00563-t009:** Partition of preservatives in water and rubber [118].

Preservative	Distribution in Water	Distribution in Rubber
Benzyl alcohol, 1%	85%	15%
Phenol, 0.5%	75%	25%
*m*-Cresol, 0.3%	67%	33%
Chlorobutanol, 0.5%	10–20%	80–90%

**Table 10 pharmaceutics-15-00563-t010:** Loss of antimicrobial preservatives (AP) from silicone tubing with an inner diameter (ID) of 1.6 mm and 6.0 mm. The partition coefficient (*k*) of each preservative was calculated. The percentage of remaining preservative in solutions were determined after incubation for 5 min and 6 h [117].

AP	*k*	ID/mm	AP Remaining after 5 min/%	AP Remaining after 6 h
*m*-Cresol	0.83	1.6	approx. 60	0
		6.0	90	34
Phenol	0.44	1.6	approx. 80	0
		6.0	97	53
Benzyl alcohol	0.27	1.6	approx. 80	0
		6.0	97	63

**Table 11 pharmaceutics-15-00563-t011:** Summary of the reported incompatibilities of antimicrobial agents with packaging material. Additionally, recommended containers and storage conditions are listed [14,117,118,127,128,129]. Abbreviation: NI = no information.

Preservative	Reported Incompatibilities with		Recommended Container
	Container/Consumables	Rubber Stopper *	
Benzalkonium Chloride	Polyvinyl chloride [14] Adsorbed by polyethylene container in the presence of counter-ions, e.g., ammonium thiocyanate [122,123]	NI	Glass or polypropylene packing materials [14,122,123] Stored in an airtight container protected from light and contact with metals [14]
Benzyl Alcohol	Methylcellulose Polyethylene [119,122,123] Silicone tubing [117]	Slowly adsorbed by natural rubber, neoprene rubber, and butyl rubber [14]	Metal and glass container [14] Polypropylene, polyvinyl chloride containers or inert containers coated with fluorinated polymers, e.g., Teflon [14,122,123] Protected from light and air [14]
Chlorobutanol	Polyethylene [122,123] Polyhydroxyethyl-methacrylate (used in soft contact lenses) [14]	Rubber stoppers [14]	Glass or polypropylene container [14,122,123] Inert container [14]
*m*-Cresol	Silicone tubing [117]	“Red rubber” * [118] “White rubber” * [118]	Well-closed container, protected from light and air [14]
Parabens	Polyethylene: adsorption increases with increasing chain length [122,123] Polyvinyl chloride [129] Silicone tubing [127]	NI	Glass container [14] Polypropylene container [127] Well-closed container [14] Fluoropolymer-based tubes [127]
Phenol	Silicone tubing [117]	“Red rubber” * [118] “White rubber” * [118]	Glass [14] Polyvinyl chloride or polypropylene container [14] Protected from light and air [14]
Phenoxyethanol	Polyvinyl chloride [14]	NI	Well-closed container [14]

* Rubber Stopper: Not Further Specified in the references.

**Table 12 pharmaceutics-15-00563-t012:** Concentration ranges of preservatives in mg/mL depending on the application route.

AP	IV	SC	IM	Nasal
Phenol	53.1	26.6–60.6	53.1	-
Benzyl alcohol	83.2–101.7	83.2–101.7	83.2–92.5	-
Chlorobutanol	28.2	-	28.2	28.2
*m*-Cresol	2.8–29.6	2.8–29.6	13.9–27.7	-
Benzalkonium chloride	-	-	-	0.28–0.568
Phenoxyethanol	-	-	43.4	-

**Table 13 pharmaceutics-15-00563-t013:** General overview of possible protein/peptide-preservative interactions and their effects. Additionally, it is shown how these effects can be counteracted.

Protein/Peptide-Preservative Interactions Can Lead to	Effect Can Be Reduced by
Negative effect on preservatives - Decreased antimicrobial activity of preservatives [19,21] - Decreased preservative mobility correlating with a preservative’s hydrophobicity [19,21] Negative effect on the API - API aggregation [5,15,16,17,19,22,23,33,145] - Partially unfolding of the API [15,16,22,23,33,145]	- Optimizing the API and preservative concentrations - Optimizing temperature conditions [16,18,23,145] - Adding preferentially excluded excipients, e.g., sucrose and trehalose, or sorbitol [16,18,23,154] - Adding co-solvents, e.g., trimethylamine N-oxide (TMAO) [155] - Choosing a pH at which charge-charge repulsion between API molecules is increased [16,23,33] - Identifying and stabilizing aggregation hotspots [22] - Including in silico models in the experimental design as those can predict physicochemical properties of the API and its binding behavior to excipients [20]

**Table 14 pharmaceutics-15-00563-t014:** Preservative-induced protein aggregation rate based on the results of previous studies.

API	Studied Protein/Peptide	Formulation Buffer	Aggregation Rate
Peptide	Palmitoylated model peptide (up to 10 mg·mL^−1^)	20 mM His/His-HCl buffer with 150 mM NaCl, pH 7.0	CR > PH > BA [19]
	Palmitoylated peptide (up to 0.17 mM)	Antioxidants, 50 mM histidine buffer, pH 7.0	CR > PH > BA [145]
Protein	Recombinant human growth hormone (10 mg·mL^−1^)	5 mM phosphate buffer, pH 7.4	CR > PH > BA [33]
	Cytochrome c (up to 3 mM)	0.1 M sodium phosphate, 0.15 M NaCl, pH 7.0	CR > PH > BA > PE > CB [22]
	Interferon α -2a (10 µM)	0.01 M ammonium acetate, 0.12 M NaCl, pH 5.0	CR > PH > BA > PE [17]
Antibody	Humanized monoclonal antibody (10 mg·mL^−1^)	Histidine buffer with PS80 and NaCl, pH 6.0	CR > PH > CB > BA > PP > MP [5]
	IgG1 mAb (1 mg·mL^−1^)	20 mM citrate—phosphate buffer, 100 mM NaCl, pH 6.0	CR > PH > PE > BA [15]

**Table 15 pharmaceutics-15-00563-t015:** Description of the multi-dose trastuzumab formulation Herceptin (Genentech, Inc., package leaflet).

Nominal Content in the Vial	
Trastuzumab	420 mg
L-histidine HCl monohydrate	9.5 mg
L-histidine	6.1 mg
α, α-trehalose dihydrate	381.1 mg
Polysorbate 20	1.7 mg
Bacteriostatic Water for Injection (BWFI)	
Benzyl alcohol	1.1%
Reconstitution with 20 mL BWFI yields to a multi-dose solution containing	
Trastuzumab	21 mg·mL^−1^
pH	Approximately 6

## Data Availability

Not applicable.

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
