# Peer review of "Antimicrobial Preservatives for Protein and Peptide Formulations: An Overview"

_pharmaceutics, 2023, doi:10.3390/pharmaceutics15020563_

Round 1

Reviewer 1 Report

General remark #1:

Although title of the article is “Antimicrobial Preservatives for Protein Formulations: An Overview”, Authors describe also studies on peptide formulations. The title should be corrected accordingly. Additionally in this case, manuscript should be implemented with definitions of (and differences between) peptide and protein drugs as well as reviewed in terms of proper reference to the type of studied molecule (as an example, chapter 3.2.1. page 69, Authors describe molecule studied by Heljo et al. (2015) as ‘acetylated model protein’ instead of ‘peptide acylated with C16’).

General remark #2:

The manuscript is very extensive. If should be divided into two parts for greater clarity. As an example, part 1 should focus on preservatives in general (chapters 2.1 – 2.3) and part 2 should focus on an overview of preservatives in protein and peptide formulations (chapters 2.4 – onwards). Alternatively, for the part 1 Authors should refer to already existing articles for general overview of preservatives used in pharmaceuticals and focus on the part 2 which is a significant contribution to the field.

General remark #3

Authors should consider implementing the manuscript with Table similar to one presented in [1] consisting of product name, presentation, dose, formulation and packaging. As Figures 1-7 suggest that Authors already did an extensive work with cataloguing the marketed products, additional table should not be connected with substantial effort. On the other hand, such database would be an valuable asset for scientists in the field.

[1] Strickley RG, Lambert WJ. A review of Formulations of Commercially Available Antibodies. J Pharm Sci. 2021 Jul;110(7):2590-2608.e56. doi: 10.1016/j.xphs.2021.03.017. Epub 2021 Mar 28. PMID: 33789155.

General remark #4

Authors should unify the units of the preservative concentration in described studies (mg/mL or % or mM) along the manuscript.

General remark #5

Authors should consider joining together chapters 2.5 and 2.6. Currently both chapters refer and describe exactly the same studies, which results in multiple repetitions and, as a consequence, decrease in the legibility of the manuscript.

Specific comments:

Comment #1

‘Second, the application is facilitated for the patient or the health staff since the same container can be used several days or weeks also providing a higher dosage flexibility’ (page 3)

Authors should clarify what they mean by ‘application is facilitated for the patient or the health staff’. The fragment should be corrected accordingly.

Comment #2

‘As described in the literature, suitable preservatives for protein formulations are m-cresol (CR), phenol (PH), benzyl alcohol (BA), benzalkonium chloride (BAK), chlorobutanol (CB), phenoxyethanol (PE), methylparaben (MP), propylparaben (PP) (…)’ (page 4)

According to Meyer et al. 2007, Phenoxyethanol is used in Vaccines. On the other hand, Authors did not mention Thiomersal.

Comment #3

‘For example, Lambert & Hammond (1973) reported that phenolic derivatives induced the efflux of potassium ions out of the microbiol cell leading to cell death (P.A. Lambert & Hammond, 1973). Benzyl alcohol, chlorobutanol and phenoxyethanol belong to the class of alcohols. This class exhibits rapid antimicrobial efficacy against a broad range of vegetative bacteria, viruses, and fungi but is not active against spores (…)’ (page 5)

Authors should unify the way they are describing the preservative groups. Only antimicrobial activity mechanisms was described for phenolic derivatives, while both antimicrobial spectrum and mechanism was presented for class of alcohols. Additionally, parabens are described only in terms of their activity spectrum. Section should be corrected accordingly.

Comment #4

‘Starting from the outside, they contain three additional layers before reaching their cell wall.’ (page 5)

Authors should clarify – Layers of what?

Comment #5

‘For example, the ranking order in which peptide - preservative interactions’ (page 7)

Authors should clarify – What type of interactions?

Comment #6

(Rowe et al., 2006); (page 102)

Authors should correct the reference.

Comment #7

‘They include the nature, composition, and condition of the microorganism’ (page 10)

Authors should clarify – What do they mean by microorganism composition?

Comment #8

‘This phenomenon is attributed to their high lipid content in their cell wall, especially the presence of mycolic acid, making them more hydrophobic’ (page 10)

Authors should clarify – More hydrophobic when compared to what? Why higher hydrophobicity results in higher resistance against preservatives?

Comment #9

Antimicrobial agents (…)’ (page 11)

Antimicrobial agent is wider term than ‘preservatives’, thus this sentence can be misleading.

Comment #10

‘Consequently, the preservative’s efficiency is directly correlated with its ability to interfere with the cell membrane or cell wall (McDonnell & Russell, 1999).’ (page 11)

1.      McDonnell & Russell, 1999 does not cover antimicrobial activity of preservatives

2.      How does this sentence correlate with the following fragment:

‘As they differ in their structural composition, they can either interact with target sites on the microbial cell surface or inside the cell (McDonnell & Russell, 1999). (page 11)’

Comment #11

‘Some preservatives are active against Gram-positive bacteria, but do not show efficiency against Gram-negative’ (page 11)

Authors should be here more specific (which preservatives?) and should add proper reference to back up this statement.

Comment #12

‘The antimicrobial efficacy of an AP can be increased by microbiologically inactive or only weakly active compounds’ (page 16)

Authors should clarify – What do they mean by ‘only weakly active compounds’?

Comment #13

‘As a chelating agent, it acts on the cell wall of microorganism (…) ‘(page 16)

Authors should clarify – How?

Comment #14

‘However, even though this effect does exist, it only occurs over a small concentration range of emulgent ‘(page 16)

Emulgent should not be used as a synonym for surfactant/ polysorbate.

Comment #15

‘Spectrophotometric analysis was used to determine the concentration of parabens, and high-performance liquid chromatography (HPLC) was used to determine the amount of preservatives in the paraben-polysorbate 80 system’ (page 17)

Authors should rephrase this sentence to clearly state why both spectrophotometric and HPLC analyses were used to determine the concentration (amount) of parabens in sample

Comment #16

‘microbistatic’ (page 18)

Authors should correct it to ‘microbiostatic’.

Comment #17

‘2.2 Antimicrobial efficacy test’ (pages 18 - 23)

Authors should consider simplifying the comparison of different compendial methods in the form of a table. Alternatively sub-chapters should be unified so the potential reader could easily compare in terms of procedure, microorganisms etc. them while reading. Lastly, Authors should refer to precise Ph. Eur. and JP chapters on antimicrobial effectiveness testing.

Comment #18

‘Phenoxyethanol shows a broad antimicrobial activity over a wide pH range (Rowe et al., 2006)’ (page 28)

Authors should precisely state the pH range.

Comment #19

‘For example, the logarithm of the preservative’s D-values for E. coli were plotted against the corresponding pH values’ (page 28)

Authors should either present such plot or not mention the procedure.

Comment #20

‘In their study, Aalto et al. (1953) described that the effectiveness of methyl-, ethyl-, propyl-, and butylparaben against A. niger did not differ from pH 4 - 7.‘ (page 28)

Authors should clarify – What does ‘did not differ from pH 4-7’ mean?

Comment #21

‘Table 6.‘ (pages 30-31)

Authors should update table with units used to describe the solubility in water and ethanol.

Comment #22

‘In a case report, a 17 year old female asthmatic patient received approximately 32 mg of benzalkonium chloride mixed in albuterol solution over 3.5 days. The multi-dose nebulizer contained 50 μg benzalkonium chloride per dose of albuterol 2.5 mg..‘ (page 32)

Authors should rephrase the sentence. Currently it might be understood as female received 32 mg of benzalkonium chloride (more than 600 doses).

Comment #23

‘However, insulin administration via continuous subcutaneous insulin infusion was re-ported to increase infection at the infusion site, and the incompatibility might be attributed to the presence of m-cresol in such formulations (Faassen et al., 1989). (page 33)

Authors should clearly state what they mean by ‘incompatibility’. Infection at infusions site should not be regarded as ‘incompatibility’.

Comment #24

‘For instance, the antimicrobial activity was reported to be considerably reduced in the presence of nonionic surfactants, e.g., polysorbate 20 (PS20) or polysorbate 80 (PS80), (page 41)

Authors should clarify – Antimicrobial activity of what preservative/s?

Comment #25

‘POE’, (page 43)

Authors should clarify – What is POE?

Comment #26

‘For example, Shi et al. (2015) studied whether m-cresol is compatible with trace levels of PS20 and PS80 focusing on whether the known incompatibility issues are dependent on the polysorbate concentration (Shi et al., 2015). They used different analytical methods including HPLC-CAD, micro-flow imaging (MFI), size exclusion chromatography (SEC), fibril count-assay, and SpectraMax M5 plate reader for turbidity measurements., (page 43)

MFI, SEC etc. were used to evaluate peptide stability when subjected to agitation, not m-cresol – polysorbate incompatibility. Authors should rephrase the section accordingly.

Comment #27

‘Thus, it might be possible that interactions with polysorbates depend on the type of preservative.’ (page 46)

Authors should consider deleting this sentence. It is an obvious statement, interactions are dependent on type of studied molecules.

Comment #28

‘In chapter 2.5.1 - 2.5.3, incompatibility issues with packaging materials are discussed..’ (page 53)

Consumables (process materials) are also described in 2.5.3.

Comment #28

‘Table 10’ (pages 54-55)

Authors should supply the table with

1)      references to each statement (e.g. None of the statements in ‘Recommended container’ column does have a reference).

2)     Explanation of abbreviation ‘NI’

3)     Correction of ‘*rubber stopper’ footnote font size

Comment #29

‘Additionally, pre-servatives are not included in infusions and freeze-dried products (Ph.Eur. 10.0, 2015; Ph.Eur. 10.0, 2018).’ (page 66)

Authors should refer to precise Ph. Eur. chapter.

Comment #30

‘Examples for physical instability are protein aggregation, surface adsorption, precipitation or denaturation (Mag-gio, 2010; Manning et al., 2010).’ (page 67)

Precipitation is not direct protein instability, but a phenomenon connected with aggregation. In other words, aggregation is condicio sine qua non of precipitation.

Comment #31

‘These sugars are discussed to be pref-erentially-excluded agents. Such excipients minimize the amount of partially unfolded protein species as they are excluded from the protein’s surface, and therefore stabilize the compact, native state (Timasheff, 2002)..’ (pages 68-69)

Sugars mechanisms of protein stabilization is that of preferential exclusion. This leads to preferential hydration of the protein as sugars are excluded from its first hydration shell. Because of this the chemical potential of the system is increased.

Comment #32

‘Hereafter, published case studies concerning the interactions of preservatives with peptides, pro-teins, and antibodies summarized to elucidate on current knowledge and approaches related to preservative interactions’ (page 69)

Authors should justify why antibodies are described separately from proteins.

Comment #33

‘Heljo et al. (2015) studied the interactions of an acetylated model protein with (…)’ and ‘The peptide had a MW of approx-imately 4.5 kDa and was acetylated at the C-terminus.’ (page 69)

Authors should correct the statement with ‘acylated model peptide’.

Comment #34

‘M-cresol showed the greatest reduction followed by phenol and benzyl alcohol.’ (page 69)

1.     Authors should rephrase this sentence – preservatives did not ‘show’ the reduction of their antimicrobial activity. Mixing them with peptide did.

2.     Authors should clarify whether phenol and benzyl alcohol mixed with peptide where characterized with the same level of reduction in antimicrobial activity when compared with protein-free samples.

Comment #35

‘Heljo et al.(2015) showed that preservative-protein interactions do not only increase tuberculin protein oligomerization(…).’ (page 70)

It is the first time Authors state that the protein evaluated in the protein is a ‘tuberculin’ protein. Referencing studied molecule in this way might confuse potential reader.

Comment #36

‘Thus, it may be concluded that phenol induced the formation of peptide species when stored in the refrigerator, but the formation is reversible. These findings did not occur in formulations including benzyl alcohol as no increase in turbidity was found even after five freeze-thaw cycles’ (page 71)

Turbidity measurement (a probabilistic evaluation of aggregate content in the sample) should not be used alone to make a statement about formation of aggregate spices.

Comment #37

‘However, as the sensitivity decreased with increasing ionic strength, it was concluded that benzyl alcohol might not interact with the peptide leading to structural changes and increased stability. (page 72)

Authors should clarify – Sensitivity of what?

Comment #38

‘Additionally, the degree of aggregation caused by agitation was consistent with the previous observations.. (page 72)

Authors should clarify – What previous observations?

Comment #39

‘these findings were in contradiction to their previous results as phenol was always reported to have a greater effect on rhGH compared to benzyl alcohol. ‘(page 72)

Authors should precisely describe the effect of phenol on rhGH (instead of greater effect).

Comment #40

‘Hence, it was concluded that the preservative induced protein aggregation of rhIL-1ra by shifting the equilibrium of the protein towards partially unfolded species. This was confirmed by ANS fluorescence analysis. It was concluded that the addition of the preserv-ative led to partial unfolding.. ‘(page 73)

Authors should rephrase those sentences as they might be hard to understand for potential reader.

Comment #41

‘as rhGCSF contains aromatic residues, ‘(page 75)

Protein containing aromatic residues is not exceptional.

Comment #42

‘To sum up, AP-induced protein destabilization and aggregation can be counteracted by (i) the ad-dition of preferentially-excluded stabilizer (e.g. sucrose, trehalose, TMAO), (ii) increasing intermo-lecular charge-charge repulsion between proteins, (iii) identifying and stabilizing aggregation hotspots, and (iv) avoiding exceedance of the temperature., ‘(page 84)

Authors should also consider adding already described selection of compatible primary packaging and process materials.

Comment #43

‘The preservative combinations are essentially determined by methylpara-bens and propylparabens ‘(page 87)

Authors should rephrase this sentence as it might be hard to understand for potential reader.

Comment #44

‘For instance, it was shown that less than 1 % high molecular weight species were formed relative to the percentage of bispecific antibody molecules in a frozen solution ‘(page 92)

Authors should rephrase this sentence as it might be hard to understand for potential reader.

Comment #45

Manuscript lacks a proper summary/conclusion at the end.

Reviewer 2 Report

1)      According to the information presented, some compounds have similar mechanisms to act as antimicrobial agents. For example, both alcohols and parabens are able to attack the membrane. On the other hand, in some parts of the text, it is said that (for example) by adding a carbon chain in parabens, their antimicrobial properties also increase. Therefore, I recommend that according to these explanations, a table should be prepared that antimicrobial compounds are classified based on the mechanism of action and those that are in the same mechanism are placed in the same group. Similarly, in the table, the factors affecting the antimicrobial properties that are encoded in their chemical structure (such as increasing the carbon chain, adding functional groups, etc.) should be expressed.

2)      Since these antimicrobial substances are included in protein drugs, it is necessary to explain their side effects after use. Although their toxicity has been well explained, no information has been mentioned about their allergenicity and immunogenicity.

3)      Please, in the interaction of proteins and antimicrobials section, provide a summary table of interactions and possible suggestions to solve these challenges. This part is very monotonous and long.

4)      Which of the compounds are currently commercialized?

5)      The text is generally well-written, but some parts need to be revised. Several examples are mentioned below:

Page 5: Microbiol?

Page 10: “However, antimicrobial activity against mycobacteria were reported” was reported

Page 11: “multiple target sites what makes” → which makes

Page 14: “disruption membrane transport” → disruption of membrane

Page 15: “was obtained for” → was obtained from

Page 15: “hydrophilic and hydrophobic phase” → phases

Page 16: “Additionally, preservatives combinations” → preservative

Page 17: “a preservatives” → preservative

Page 19: “filtration or plate count method” → filtration and/or plate count methods

Page 23: “Tests of the JP is” → are

Page 37: “were reduced compared” → was

Page 37: “All death neonates” → dead

Page 37: “showed increased”→ an increased

Page 60: “it was is” → delete “is”

Page 74: “benzyl alcohol solutions → solution

Page 75: “different analysis” → analyses

Page 77: “in accordance to → with

Page 78: “still high with 91.3” → at

Page 79: “either stored at either” → delete first either

Page 79: “two weeks storage” → two weeks of storage

Page 79: “Neither smaller bands or” → nor  

Author Response

"Please see the attachment

Round 2

Reviewer 1 Report

Authors did improve the manuscript significantly.

Nevertheless:

Addendum to Comment #10

‘Consequently, the preservative’s efficiency is directly correlated with its ability to interfere with the cell membrane or cell wall (McDonnell & Russell, 1999).’ (page 11)

1.      McDonnell & Russell, 1999 does not cover antimicrobial activity of preservatives

2.      How does this sentence correlate with the following fragment:

‘As they differ in their structural composition, they can either interact with target sites on the microbial cell surface or inside the cell (McDonnell & Russell, 1999). (page 11)’

 [1] On p. 236, chapter 11 of McDonnel & Russel, 1999, it is stated that “Before an antibacterial agent can exert its effect on a cell it must combine with that cell. This process often follows the pattern of an adsorption isotherm. […] The possession of surface activity per se may be an important factor in the antibacterial action of a group of drugs, for example the cationic detergents. The addition of low concentrations of surface-active compounds may potentiate the biological effect of an antibacterial agent.” Therefore, I’d argue that the citation is correct.

[2] “Either” was eliminated out of the sentence. Some preservatives have intracellular and membrane targets (Fraise et al. (2013)).

Authors should refer to ‘Pharmaceutical Microbiology’ by Hugo & Russell, instead of article by McDonnel & Russel in this case.

Comment #30

‘Examples for physical instability are protein aggregation, surface adsorption, precipitation or denaturation (Mag-gio, 2010; Manning et al., 2010).’ (page 67)

Precipitation is not direct protein instability, but a phenomenon connected with aggregation. In other words, aggregation is condicio sine qua non of precipitation.

Precipitation is related to phase separation of the native protein, whereas aggregation is more related to the formation of higher molecular weight species, protein particle formation, of non-native units. Not to mix up with the term “association”

Precipitation is phase separation of aggregated protein regardless if the aggregation occurred through native or non-native protein species.

According to [1]:

'Although growth occurs initially via addition of aggregation-prone monomers, subsequent growth becomes dominated by aggregate-aggregate coalescence and even phase separation of the previously formed aggregates'

'Phase-separation or precipitation of proteins can also occur without a significant conformational change (e.g., “salted out” native aggregates).'

[1] Roberts, Christopher J. “Therapeutic protein aggregation: mechanisms, design, and control.” Trends in biotechnology vol. 32,7 (2014): 372-80. doi:10.1016/j.tibtech.2014.05.005

Author Response

Answers to Reviewer Comments and Suggestions – Round 2

Reviewer 1

Comments and Suggestions for Authors

Authors did improve the manuscript significantly.

Nevertheless:

Addendum to Comment #10

‘Consequently, the preservative’s efficiency is directly correlated with its ability to interfere with the cell membrane or cell wall (McDonnell & Russell, 1999).’ (page 11)

  1. McDonnell & Russell, 1999 does not cover antimicrobial activity of preservatives
  2. How does this sentence correlate with the following fragment:

‘As they differ in their structural composition, they can either interact with target sites on the microbial cell surface or inside the cell (McDonnell & Russell, 1999). (page 11)’

 [1] On p. 236, chapter 11 of McDonnel & Russel, 1999, it is stated that “Before an antibacterial agent can exert its effect on a cell it must combine with that cell. This process often follows the pattern of an adsorption isotherm. […] The possession of surface activity per se may be an important factor in the antibacterial action of a group of drugs, for example the cationic detergents. The addition of low concentrations of surface-active compounds may potentiate the biological effect of an antibacterial agent.” Therefore, I’d argue that the citation is correct.

[2] “Either” was eliminated out of the sentence. Some preservatives have intracellular and membrane targets (Fraise et al. (2013)).

Authors should refer to ‘Pharmaceutical Microbiology’ by Hugo & Russell, instead of article by McDonnel & Russel in this case.

Reference adjusted to ‘Pharmaceutical Microbiology’ by Hugo & Russell.

Comment #30

‘Examples for physical instability are protein aggregation, surface adsorption, precipitation or denaturation (Mag-gio, 2010; Manning et al., 2010).’ (page 67)

Precipitation is not direct protein instability, but a phenomenon connected with aggregation. In other words, aggregation is condicio sine qua non of precipitation.

Precipitation is related to phase separation of the native protein, whereas aggregation is more related to the formation of higher molecular weight species, protein particle formation, of non-native units. Not to mix up with the term “association”

Precipitation is phase separation of aggregated protein regardless if the aggregation occurred through native or non-native protein species.

According to [1]:

'Although growth occurs initially via addition of aggregation-prone monomers, subsequent growth becomes dominated by aggregate-aggregate coalescence and even phase separation of the previously formed aggregates'

'Phase-separation or precipitation of proteins can also occur without a significant conformational change (e.g., “salted out” native aggregates).'

[1] Roberts, Christopher J. “Therapeutic protein aggregation: mechanisms, design, and control.” Trends in biotechnology vol. 32,7 (2014): 372-80. doi:10.1016/j.tibtech.2014.05.005

Thank you very much for this comment. We agree that precipitation is not only related to phase separation of native proteins. However, according to [145] precipitation can either result from large aggregates or from structurally unaltered protein exceeding its solubility limit. While the latter process is reversible and the protein retains its activity, the first process – precipitation following aggregation – is irreversible. For this review, we follow the definition given in [145].

  1. Manning, M.C.; Chou, D.K.; Murphy, B.M.; Payne, R.W.; Katayama, D.S. Stability of Protein Pharmaceuticals: An Update. Pharmaceut Res 2010, 27, 544–575, doi:10.1007/s11095-009-0045-6.

‘Conversely, there are physical instabilities for proteins in which the chemical composition is unaltered, but the physical state of the protein does change. This includes denaturation, aggregation, precipitation, and adsorption (Table I). The term precipitation is used here to denote insolubility rather than insoluble aggregate formation. […] For the purposes of this review, aggregation is restricted to formation of soluble aggregates, where precipitation refers to a macroscopic event where the protein can be seen coming out of solution. As seen below, precipitation may or may not be connected with aggregation. It may simply be due to conditions whereby the protein has exceeded its solubility limit. […] On the other hand, not all insoluble protein material is due to aggregation. One could have a protein that is salted out, that is, the addition of an excluded solute has caused the chemical potential of the protein to exceed that of the solid phase (326,327). While our understanding of protein solubility is still imperfect, there have been significant advances in the past 20 years (326,328). Salted-out proteins still retain activity and native-like structure (327,329–331), and the precipitation is fully reversible upon dilution.’

For the sake of clarity, we have rephrased this sentence as follows:

‘Examples for physical instability are protein aggregation, surface adsorption, denaturation, or precipitation; the latter one can result from protein aggregate growth, but also from proteins being salted out, e.g., by an excluded solute [145,146].’

Reviewer 2

No Comments and Suggestions to Authors